# Molecular characterisation of the *Bacillus subtilis* SpbK antiphage defence system

Biswa P. Mishra[1], Christian L. Loyo[2], Yanyao Cai[3], Thomas Litfin[4], Gause Miraj[1], Lou Brillault[5], Veronika Masic[1], Tamim Mosaiab[1], Premraj Rajaratnam[1], Santosh Rudrawar[1,6], Weixi Gu[7], Bostjan Kobe [7], Joseph P. Gerdt [3], Alan D. Grossman [2], Yun Shi [1,8] ✉ & Thomas Ve [1] ✉

Bacteria have a variety of mechanisms for limiting predation by phages. SpbK is a Toll/interleukin-1 receptor (TIR) domain-containing antiphage defence protein from *Bacillus subtilis* that provides protection against the temperate phage SPβ via abortive infection. Here we structurally characterise SpbK and its interaction with the SPβ protein YonE. We demonstrate that SpbK is an NADase that produces both ADP-ribose (ADPR) and canonical cyclic ADPR with a N1-glycosidic bond (cADPR, also referred to as N1-cADPR). Combining cryo-EM, in silico predictions, site-directed mutagenesis, and phage infection assays, we show that formation of two-stranded head-to-tail assemblies of SpbK TIR domains is required for both NADase activity and antiphage defence. We also demonstrate that YonE is a dodecameric portal protein that activates the NADase function of SpbK by facilitating TIR domain clustering. Collectively, our results provide insight into how bacterial TIR NADases recognise phage infection.

Bacteria have highly diverse and sophisticated immune systems to protect themselves from predation by phages. Restriction modification and CRISPR-Cas systems are well known, but over the past decade, many new defence mechanisms have been described. These include immune systems that disrupt membrane integrity[1,2], produce small molecule inhibitors of phage propagation[3–5], rely on reverse transcription of small RNAs[6–8] or proteolytic cleavage[9], modify nucleotides and viral DNA[10,11], and deplete bacteria of nucleotides required for phage replication[12,13] or energy metabolism[14–19].

A commonly used strategy in many of these immune systems is abortive infection, in which the infected cell commits suicide before the phage can complete its replication cycle, thereby protecting the bacterial community by preventing the spread of phages to neighbouring cells[20]. Several bacterial immune systems, including Thoeris,

CBASS (cyclic-oligonucleotide-based antiphage signaling system), Pycsar (pyrimidine cyclase system for antiphage resistance), Avs (antiviral STANDs), DSR (defence-associated sirtuin), SIR2-HerA, and SPARTA (short prokaryotic argonaute TIR-APAZ), have been shown to trigger abortive infection by depleting the cells of the essential energy metabolite nicotinamide adenine dinucleotide (NAD+)[14,16–19,21,22]. These systems feature either SIR2 (silent information regulator 2) or TIR domains that can rapidly hydrolyse NAD+ into nicotinamide (NAM) and ADPR.

TIR domains are found in animal, plant, and bacterial immune systems, and function through self-association and homotypic interactions with other TIR domains[23,24]. They were first described as protein-protein interaction modules mediating signalling downstream of the Toll-like receptor (TLR) and interleukin-1 receptor families in

[1]Institute for Biomedicine and Glycomics, Griffith University, Gold Coast, QLD, Australia. [2]Department of Biology, Massachusetts Institute of Technology, Cambridge, MA, USA. [3]Department of Chemistry, Indiana University, Bloomington, IN, USA. [4]Structural Biology Facility, Mark Wainwright Analytical Centre, University of New South Wales, Sydney, NSW, Australia. [5]Centre for Microscopy and Microanalysis, University of Queensland, St Lucia, QLD, Australia. [6]School of Pharmacy and Medical Sciences, Griffith University, Gold Coast, QLD, Australia. [7]School of Chemistry and Molecular Biosciences, Institute for Molecular Bioscience and Australian Infectious Diseases Research Centre, University of Queensland, St Lucia, QLD, Australia. [8]Present address: Medicinal Chemistry, Monash Institute of Pharmaceutical Sciences, Monash University, Parkville, VIC, Australia. ✉e-mail: yun.shi1@monash.edu; t.ve@griffith.edu.au

animals[25], where they induce signalling via higher-order oligomerisation[26–30]. However, studies of the neurodegenerative disease therapeutic target SARM1 (sterile alpha and TIR motif containing 1), plant immune receptors, and many bacterial TIR domain-containing proteins revealed that TIR domains have self-association-dependent NADase activity[15,17,31–35]. SARM1 is the founding member of the TIR NADase family and catalyses the hydrolysis of NAD⁺ to produce ADPR and N1-cADPR[33]. This activity leads to rapid NAD⁺ depletion, triggering axon degeneration. In bacterial immune systems, TIR domain NADases can either serve as effectors that rapidly deplete cellular NAD⁺ or producers of immune signalling molecules such as histidine conjugated ADPR (His-ADPR)[36,37], and the cADPR isomers 3′cADPR[18,32,38], 2′cADPR[32,39] and N7-cADPR[9], that differ from N1-cADPR in the nature of the cyclic linkage between the anomeric position of the distal ribose and the adenosine moiety (Fig. S1a).

Many strains of *Bacillus subtilis* are lysogenic for the temperate phage SPβ[40]. Additionally, the integrative and conjugative element ICE*Bs1* encodes a TIR domain NADase, SpbK, that is necessary and sufficient for protection against SPß[41]. The anti-SPß phenotype and NADase activity of SpbK are activated by the product of the SPß gene *yonE*. YonE is homologous to portal proteins like Gp6 of phage SPP1[42]. Portal proteins are required for procapsid assembly and correct DNA packaging and are thus essential for phage production [41].

SpbK causes depletion of NAD⁺ in *B. subtilis* cells via interaction with YonE[43]. Here, we present a structural analysis of the SpbK antiphage defence system. We show that SpbK has self-association-dependent NADase activity and produces both ADPR and N1-cADPR. SpbK exists as a stable dimer in solution that can further self-associate into filaments. A 3.3 Å resolution cryoEM structure of the filament revealed a two-stranded higher-order assembly arrangement of SpbK TIR domains that resembles the TIR domain assembly formed by STING-TIR[41]. Mutagenesis studies demonstrate that the interfaces within these assemblies are required for both SpbK NADase activity and antiphage defence. We also determined the cryoEM structure of dodecameric YonE at 3.3 Å resolution and found that YonE interacts directly with SpbK and activates its NADase function. AlphaFold modelling combined with mutagenesis and gel-filtration assays further demonstrate that the clip domains in the dodecameric portal serve as docking stations for the N-terminal domain of SpbK, driving SpbK clustering and activation of its NADase function. Collectively, our studies provide a detailed structural model for SpbK activation by YonE.

## Results

### SpbK is an NADase that produces ADPR and N1-cADPR
SpbK consists of an N-terminal 4-helix bundle domain (residues 1–94; SpbK^Ntd), an 18-residue flexible linker, and a C-terminal TIR domain (residues 113–266; SpbK^TIR) (Fig. 1a). We expressed and purified full-length SpbK as an MBP (maltose-binding protein) fusion protein (MBP-SpbK). Real-time NMR-based NADase assays confirmed that SpbK is enzymatically active and produces NAM, ADPR and N1-cADPR (Figs. 1b, c and S2). Similar to N1-cADPR produced by the mammalian NADase CD38[44], N1-cADPR produced by SpbK was unstable in the reaction mixture and eventually converted to ADPR. Consistent with the enzymology of TIR domain NADases[32,37,45], SpbK also catalysed base-exchange reactions with various heterocyclic bases, including imidazole, IP6C[46], 5-iodo-isoquinoline (1), 1,2-dihydro-2,7-naphthyridin-1-one (2), and 8-amino-isoquinoline (3) (Fig. S1b, c).

### N1-cADPR is produced in SpbK-containing cells after SPβ phage infection and YonE expression
Since N1-cADPR was not previously reported as a product of SpbK in cells, we asked whether it was an artefact of the in vitro assay or if N1-cADPR is actually produced in SpbK-expressing cells following infection with SPβ. We examined the cell lysate by liquid chromatography–tandem mass spectrometry (LC-MS/MS). Cells harbouring SpbK generated a peak with m/z consistent with N1-cADPR (Fig. 2a). The same peak was observed in cells expressing both YonE and SpbK, in the absence of phage infection (Fig. 2b). As observed for N1-cADPR in vitro, these peaks were short-lived. The MS/MS fragmentation pattern of these peaks matched that of N1-cADPR (Fig. 2c). Unexpectedly, the material eluted as two peaks with identical m/z and fragmentation patterns (Fig. 2a–c). We hypothesized that either two cADPR variants were produced by the cell, or components in the cell lysate (e.g., divalent metal cations) caused the N1-cADPR to adopt different configurations with different retention times. To discriminate between these possibilities, we added purified N1-cADPR to the cell lysate that had been filtered to remove >3 kDa components and lacked endogenous N1-cADPR. If components in the lysate cause different configurations to form, then this exogenous N1-cADPR should elute as two peaks. Indeed, we observed two peaks (Fig. 2d). Therefore, we concluded that an unknown component of the cell lysate influences the migration of the analyte, splitting it into two separate peaks. As further confirmation of this hypothesis, we pre-purified the lysate before running on LC-MS/MS and found that the two peaks converted to one (Fig. 2e). Importantly, co-injection of the pure N1-cADPR standard with this purified induced cell lysate revealed no new peaks, indicating that the observed peaks matched N1-cADPR (Fig. 2e).

### SpbK assembles into filaments
We found that MBP-SpbK exists as a dimer in solution (Fig. 3a, Table S2). SpbK^Ntd is monomeric (Fig. 3b, Table S2), suggesting that the TIR domain is required for MBP-SpbK dimerisation. We also observed that MBP-SpbK solutions become turbid when the MBP-tag is removed via TEV protease cleavage, suggesting higher-order oligomerisation. Sodium dodecyl sulphate–polyacrylamide gel electrophoresis (SDS-PAGE) analysis revealed that more than 90% of SpbK was present in the insoluble fraction when MBP-SpbK (40 µM) was incubated with TEV protease (Fig. 3c). Negative-stain electron microscopy (EM) of MBP-SpbK incubated with TEV protease revealed the presence of long filaments with a diameter of 15–20 nm (Fig. 3d). We also observed that MBP-SpbK has significantly higher NADase activity in the presence of TEV protease (Fig. 3e), indicating that filamentation enhances this activity.

### The cryoEM structure of SpbK reveals the molecular mechanism underlying its self-association-dependent NADase activity
We solved the structure of the SpbK filament using cryoEM (Figs. 4a, b and S3; Table S3). The three-dimensional reconstruction of the filament, at an overall resolution of 3.3 Å, has D1 symmetry, a helical rise of 8.9 Å and twist of 79.3° per subunit. The filament is a hollow tube with outer and inner diameters of ~210 Å and ~160 Å, respectively (Fig. 4b). The SpbK cryoEM density map allowed us to build a model covering residues 111-266, which correspond to the TIR domain of SpbK (Fig. 4c, d). No density was observed for the N-terminal domain and linker region, which presumably adopts a dynamic conformation. Each protomer of the filament shows the canonical TIR domain structure, consisting of five β-strands surrounded by five α-helices; the loops are named based on the elements of secondary structure they connect (Fig. 4d). The TIR domain of SpbK also has an additional helix, αF, that folds back onto the surface region encompassing the αA and αE helices. Among the known TIR domain structures, *Maribacter polysiphoniae* SPARTA (PDB: 8SPO) displays the highest similarity to SpbK (RMSD of 2.6 Å for 133 Cα atoms) (Fig. 4d).

The SpbK filament is composed of five protofilaments; each having two antiparallel strands of TIR domains in a similar arrangement to the SfSTING filament[47] (Fig. 4c, e). The SpbK protofilaments have two major interfaces (Fig. 4f). The first interface is the intrastrand head-to-tail "BE" interface, with a buried surface area of 771 Å². The BE interface is asymmetric and involves residues in the BB loop of one protomer

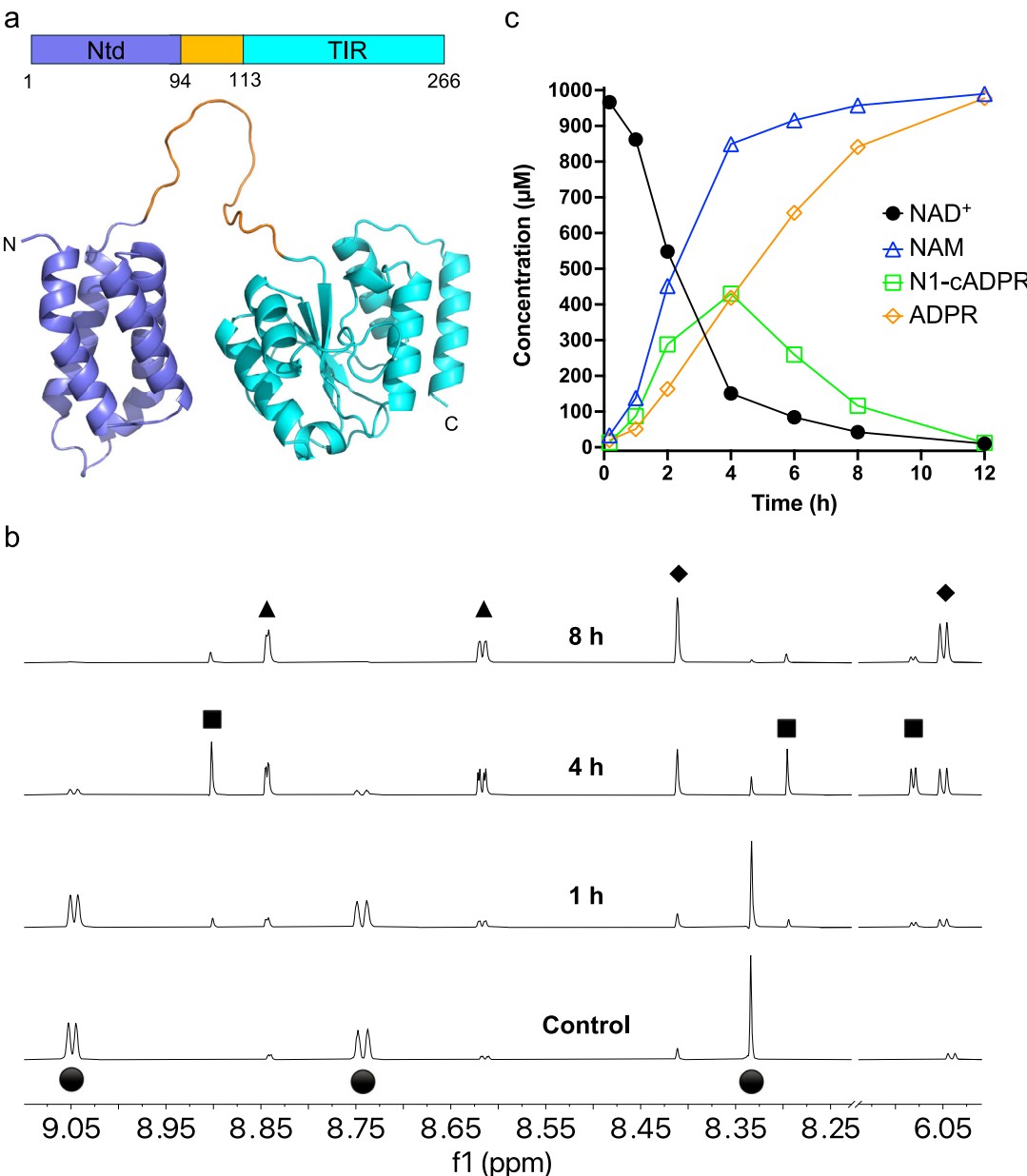

**Fig. 1 | SpbK is an NADase that produces ADPR and N1-cADPR. a** Schematic representation and AlphaFold 3 model of SpbK. The N-terminal domain, linker and the C-terminal TIR domain are coloured in slate, orange and cyan, respectively. **b** Expansions of ¹H NMR spectra showing NADase activity for SpbK. The initial NAD⁺ concentration was 1 mM, and the protein concentration was 5 µM. Spectra correspond to 1 h, 4 h, and 8 h incubation time, except for the control. Selected peaks are labelled, showing the production of nicotinamide (NAM) (black triangles), ADPR (black diamonds), and N1-cADPR (black squares) from NAD⁺ (black circles). **c** Reaction progress curves for SpbK-catalysed reaction in (**b**).

(TIR-A) interacting with residues of the DD loop, βE strand and αE-helix of the other (TIR-B) protomer (Fig. 4f). At the core of this interface, D158 of the BB loop forms hydrogen bonds with S204 in the βD strand and K240 in the αE-helix. The interface is also stabilised by three main chain hydrogen bonds (D158-K224, E159-K224 and I161-N222). This head-to-tail association creates the composite active site of TIR NADases, consisting of two TIR subunits (TIR-A and TIR-B)[26,32,45]. An AlphaFold 3 model of a SpbK:NAD⁺ hexamer is almost identical to the protofilament structure (Fig S3e–g), and the NAD⁺ binding mode is similar to that of **1AD** and **3AD** in complex with SARM1ᵀᴵᴿ and AbTirᵀᴵᴿ, respectively[32,45]. The model suggests that residues S121, S122, T148, I155, I161, F162, L165, Y188, and E192 in TIR-A may be involved in coordinating binding to the nicotinamide base, the nicotinamide ribose and the diphosphate group of NAD⁺ (Fig. 4g). In the SARM1ᵀᴵᴿ

and AbTirᵀᴵᴿ complex structures, the adenine base forms interactions with a tryptophan side chain in TIR-B. There are no aromatic residues in this region of SpbK TIR-B, but the AlphaFold 3 model of the SpbK:NAD⁺ hexamer suggests that K224 may be involved in the interaction with the adenine base.

The second interface is the interstrand "CC" interface between the two strands of the protofilament, with a buried interface area of 1902 Å² (Fig. 4f). Like the SfSTING and one type of MyD88 filaments[47,48], this interface is symmetric and highly conserved residues in the αC helix (N191, G194 and W197) form extensive packing interactions. The SpbK interstrand interaction involves only one molecule from each strand and does not contribute to stabilising the intrastrand BE interface. This contrasts with the SARM1ᵀᴵᴿ and AbTirᵀᴵᴿ assemblies, where the interstrand interface involves one molecule engaging with two

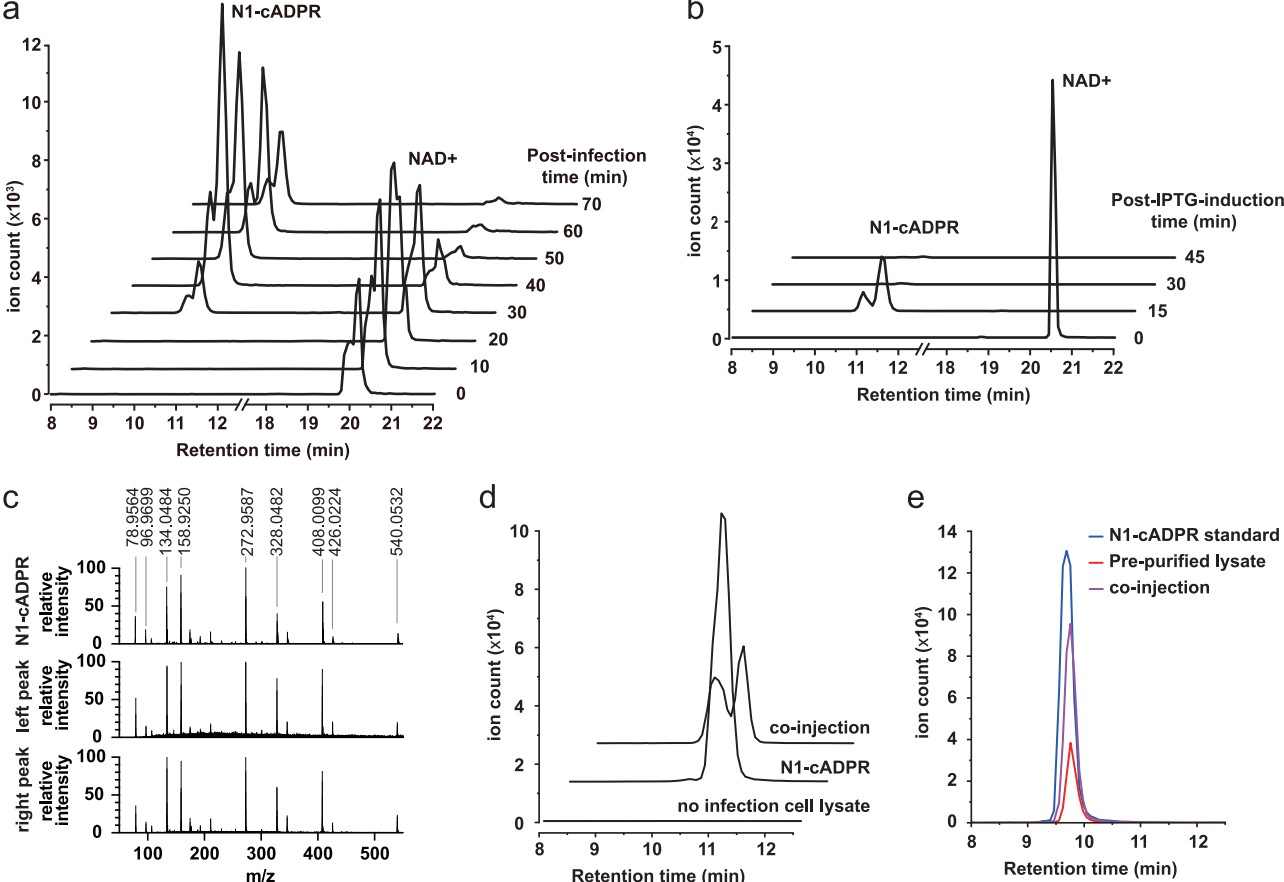

**Fig. 2 | N1-cADPR production is induced by SpBeta infection and YonE expression in SpbK-containing cells. a** Extracted ion chromatograms (EICs) for m/z 540.0431–540.0649 (matching N1-cADPR [M−H]−) and m/z 662.0887–662.1153 (matching NAD+ [M−2H]−) in the lysate of *B. subtilis* cells containing SpbK, at 10-minute intervals following infection with SpBeta clear-plaque phage. **b** EICs for m/z 540.0431–540.0649 (matching N1-cADPR [M−H]−) and m/z 662.0887–662.1153 (matching NAD+ [M−2H]−) in the lysate of *B. subtilis* cells containing SpbK, at 15-minute intervals following induction of YonE expression with IPTG. **c** MS/MS spectra for the N1-cADPR standard (top), left N1-cADPR peak from the lysate of cells 40 minutes after phage infection (middle), and right N1-cADPR peak from the lysate of cells 40 minutes after phage infection (bottom), reveal that both N1-cADPR peaks have identical MS/MS spectra with N1-cADPR. **d** Co-injection of purified N1-cADPR with lysate from non-induced cells that lack N1-cADPR (top), N1-cADPR standard injected in water (middle), and plain lysate from non-induced cells that lack N1-cADPR (bottom), reveals that cell lysate causes N1-cADPR to split into two peaks under our HPLC conditions. **e** Co-injection of N1-cADPR standard with purified lysate from SpbK/YonE cells (purple), purified lysate alone (red), and standard alone (blue), reveal that the SpbK-produced metabolite co-elutes with N1-cADPR.

molecules from the opposite strand, thereby stabilising the intrastrand BE interface (Fig. 4e).

The interfaces between the SpbK protofilaments involve residues predominantly located in the αE and αF helices ("EF" interface) and have a buried surface area of 774 Å² (Fig. 4f). We speculate that these inter-protofilament interactions are most likely equivalent to non-biological crystal contacts[27,30,49].

Overall, our structural data suggest that, like other TIR NADases such as SARM1, AbTIR, STING-TIR, SAVED-TIR, and plant NLRs[15,32,45,47,50,51], SpbK TIR domains form two-stranded assemblies with composite active sites at the intrastrand BE interfaces.

## SpbK filamentation controls NADase activity and antiphage defence

Using site-directed mutagenesis, we made altered residues in the interstrand and intrastrand interfaces. The interstrand αC helix mutants N191R, G194R, and W197A were defective in forming filaments based on negative-stain EM of all three mutants (Fig. 5a, S4a). Mass photometry and MALS analyses also showed that the N191R and W197A mutants were monomeric while the G194R mutant formed both monomers (P1) and dimers (P2) (Figs. 5b and S4b, Table S2). Importantly, these three mutants were also defective in NADase activity (Fig. 5c). The D158R intrastrand BB loop mutant formed stable dimers

and assembled into filaments, although these were shorter than those of wild-type SpbK (Fig. 5a). This mutant was also defective in NADase activity (Fig. 5c), indicating that the D158R substitution perturbs the intrastrand interface, resulting in an alternative mode of oligomerisation. This alteration likely compromises the structural integrity of the active site, which resides at the intrastrand BE interface, thereby abrogating its capacity to bind and hydrolyse NAD+. We also tested the D158R, N191R, G194R, and W197A mutants in *B. subtilis*. All four mutations abolished antiphage defence against SPβ phages (Fig. 5d–g). SPβ was able to grow and make plaques on cells expressing these *spbK* mutants, in contrast to the absence of phage growth in cells expressing wild-type *spbK* (Fig. 5d, e). All of the mutants exhibited growth comparable to wild-type *spbK* (Fig. 5f), and consistent with their loss of NADase activity, they did not induce growth arrest in the presence of YonE (Fig. 5g). Collectively, our mutagenesis data demonstrate that the observed arrangement of TIR domains within the SpbK filament is important for both its catalytic activity and phage defence, and that an intact BB loop is required.

## YonE is a dodecameric portal protein

The SPβ phage protein YonE activates the antiphage defence activity of SpbK and is predicted to be a portal protein[43]. We expressed and purified a truncated YonE construct lacking the first 45 residues

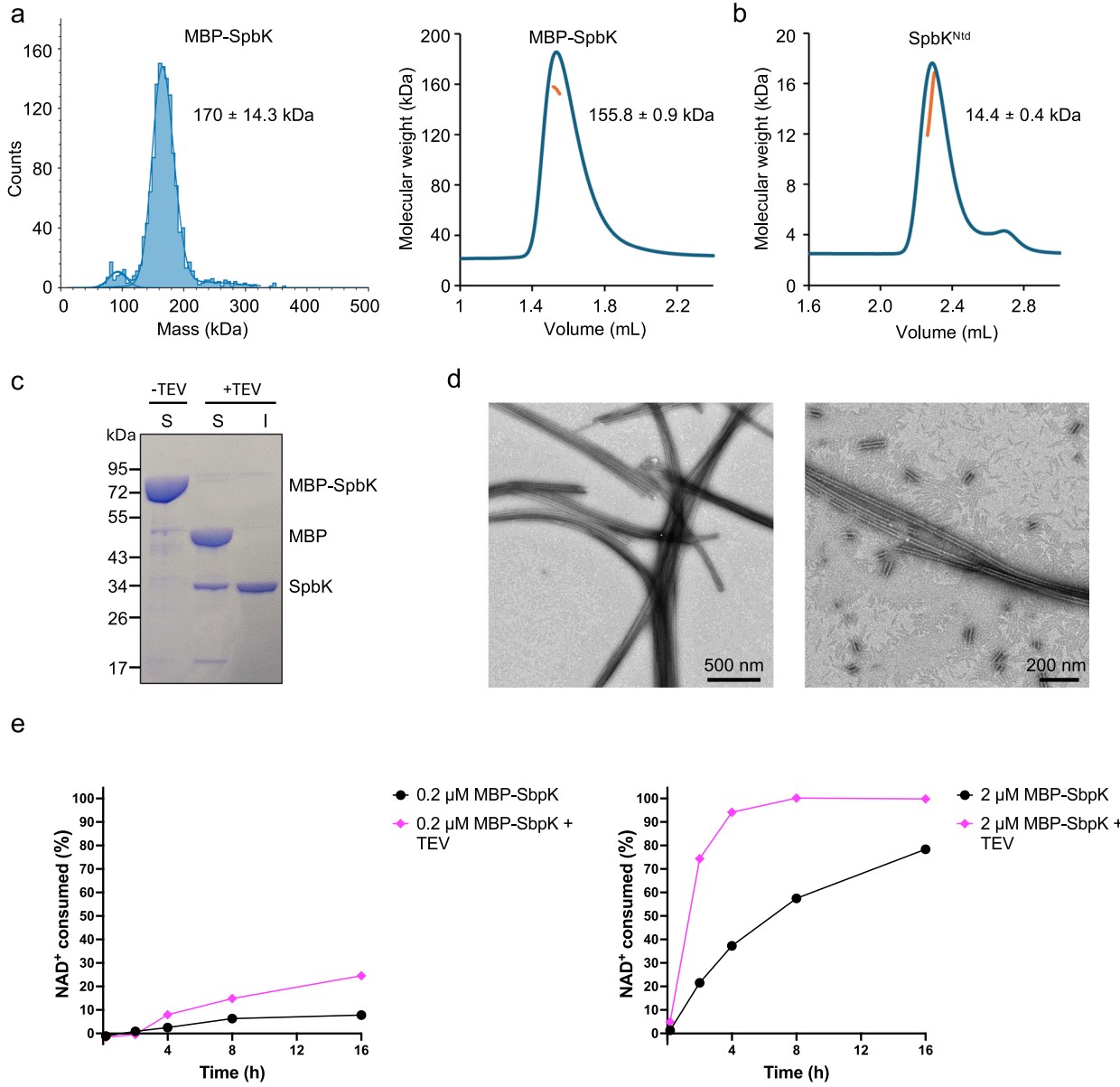

**Fig. 3 | SpbK oligomerisation. a** Mass photometry and size-exclusion chromatography–coupled multiangle light scattering (SEC-MALS) analysis of MBP-SpbK. Mass photometry experiments were conducted three times with similar results. **b** SEC-MALS analysis of SpbK$^{Ntd}$. The blue line represents the refractive index trace, while the orange line represents the average molecular mass distribution across the peak. **c** SDS-PAGE analysis of the soluble (S) and insoluble (I) fractions after incubation of MBP-SpbK (40 µM) with TEV protease at room temperature for 2 h. Experiments were repeated three times with similar results. **d** Representative negative-stain EM images of MBP-SpbK incubated with TEV protease. Micrographs were recorded from five grids from five independent samples. **e** Reaction progress curves of 0.2 µM MBP-SpbK ±TEV + 500 µM NAD$^+$ (left panel) and 2 µM MBP-SpbK ±TEV + 500 µM NAD$^+$ (right panel). MBP-SpbK was incubated with TEV for 1 hr before the addition of NAD$^+$. The experiments were conducted twice with similar results.

predicted to be disordered[52]. Negative-stain EM revealed ring-shaped YonE particles with a diameter of 20 nm (Fig. 6a), and mass photometry analysis showed that YonE exists as a dodecamer in solution (Fig. 6b). Initial cryoEM analysis confirmed that YonE forms a dodecameric assembly featuring a central channel characteristic of portal proteins (Fig. S5a). However, the dataset exhibited preferential particle orientation, which prevented detailed structure determination. To address this limitation, we acquired an additional dataset using a 30° tilt angle. Merging the tilted and untilted datasets led to a reconstruction at an overall resolution of 3.3 Å (Figs. 6c and S5b–e). The resulting cryoEM map, although moderate in quality, allowed us to build a model encompassing residues 46–248 and 369–466 of YonE (Fig. 6d). These regions correspond to the wing (residues 60–240 and 369–414) and crown (residues 415–465) domains. Densities for the

stem and clip domains and the tunnel loop (residues 249–368) were absent. We speculate that this lack of density arises from both the preferred orientation issue and increased flexibility of the stem and clip domains when YonE is not embedded within the procapsid or associated with the terminase or tail machinery.

To gain structural insight into the missing regions, we modelled the dodecameric assembly using AlphaFold 2 (Figs. 6e and S6). The predicted wing and crown domains closely match the cryoEM structure. The stem domain comprises two long helices connected by the clip domain, which consists of three β-strands and one α-helix (Fig. 6e), forming the base of the portal assembly. The model suggests that all four subdomains (wing, stem, clip, and crown) contribute to intersubunit interactions within the dodecamer. According to DALI structure comparison calculations[53], close structural relatives include portal

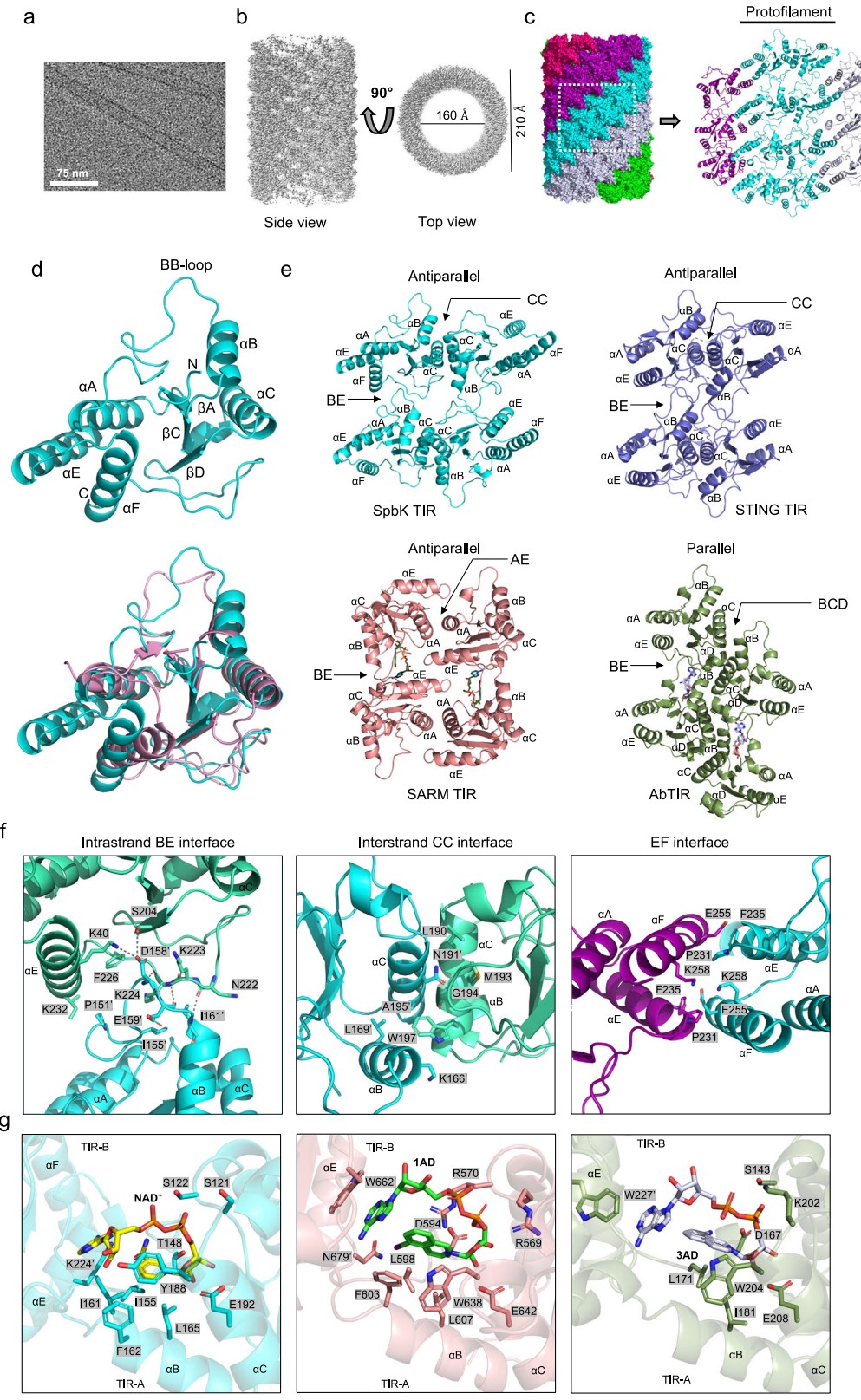

proteins from the PhiT myophage, siphophage lambda and bacteriophage G20C (Fig. 6f).

## YonE activates the NADase function of SpbK

Incubation of MBP-SpbK with YonE resulted in significantly enhanced NADase activity (Fig. 7a), demonstrating that YonE interacts directly with SpbK to activate its NADase function. The D158R intrastrand

interface mutant and the N191R, G194R, and W197A interstrand interface mutants did not display NADase activity in the presence of YonE, suggesting that YonE-induced SpbK clustering cannot overcome the destabilising effects of these mutants on TIR domain oligomerisation (Fig. S7a).

Loyo and Grossman recently reported that the N-terminal domain of SpbK is required for co-immunoprecipitation of YonE and SpbK

**Fig. 4 | CryoEM structure of the SpbK filament. a** Representative cryoEM image of SpbK filaments. 11,993 micrographs were collected from one grid preparation. **b** Electrostatic potential density map of the SpbK filament. **c** Surface representation of the filament. The 5 protofilaments are coloured red, magenta, cyan, light-blue and green, respectively. **d** Top: cartoon representation of an individual SpbK TIR subunit. Bottom: Structural superposition of SpbK (cyan) and *Maribacter polysiphoniae* SPARTA (PDB: 8SPO; pink) TIR domains. **e** Comparisons of SpbK, STING-TIR (PDB: 7UN8), SARM1[TIR]:1AD (PDB:7NAK) and AbTir[TIR]:3AD (PDB: 7UXU) assemblies. BE interface: BB loop, βD and βE strands and αE-helix; BCD interface: αB, αC,

αD helices and CD loop; AE interface: αA and αE helices and EE loop; CC interface: αC helix. In the AbTir[TIR] assembly, the two strands are offset, and the aC helix interacts with two TIR domains from the opposing strand. In SARM1, the EE loop engages with two TIR domains in the second strand, thereby stabilising the intrastrand BE interface. **f** Detailed interactions within the SpbK filament. Left: BE interface; Middle: CC interface; Right: EF interface. For the symmetric CC interface, only one set of the interacting residues is highlighted. **g** Comparison of active site region in SpbK AlphaFold 3 model, SARM1[TIR] (PBD:7NAK) and AbTir[TIR] (PDB: 7UXU).

from *B.subtilis* lysate[43]. They also found that a YonE clip domain mutant (A291P), which facilitates SPβ escape, did not co-immunoprecipitate with SpbK. We used analytical gel-filtration assays to confirm that YonE interacts directly with SpbK[Ntd] (Fig. 7b and S7b, c). Because SpbK exists as a dimer in solution, we tested SpbK[Ntd] by itself and fused to glutathione S-transferase (SpbK[Ntd]-GST) and the tandem sterile alpha motif (SAM) domains of human SARM1 (SpbK[Ntd]-hSARM[SAM]). GST and hSARM1[SAM] exist as a stable dimer and octamer, respectively[31,54]. SDS-PAGE analysis of gel-filtration fractions shows that all three variants of the SpbK N-terminal domain co-elute with YonE. We also expressed and purified the YonE A291P mutant. MBP-SpbK incubated with YonE A291P exhibits weak NADase activity, compared to MBP-SpbK incubated with wild-type YonE (Fig. 7c). The A291P substitution also weakened the interaction with SpbK[Ntd]-hSARM[SAM] (Fig. 7d). These results are consistent with Loyo and Grossman's observation that the A291P mutant only partially escapes SpbK-mediated defence[43]. Together, these results demonstrate that YonE activates SpbK by interacting directly with its N-terminal domain.

To provide structural insight into how SpbK[Ntd] interacts with YonE, we modelled their interaction using AlphaFold 3 (Figs. 7e, f and S8). In the model, SpbK[Ntd] interacts with the clip domains of YonE and forms a dodecameric ring at the base of the portal (Fig. 7e). The YonE clip domain helix packs against the α3-α4 loop and the end of the α1 helix in SpbK[Ntd]. Both F69 and F71 in the α3-α4 loop of SpbK[Ntd] form extensive hydrophobic contacts with the clip domain (Fig. 7f). The SpbK[Ntd] molecules also form asymmetric interactions with each other; the α1 and α4 helices of one protomer interact with the α2 and α3 helices of the other protomer (Fig. 7f). A291 is located in the clip domain helix of YonE, and substitution of this residue with proline is likely to induce structural changes in this helix and alter interactions with SpbK[Ntd], consistent with our NADase and gel-filtration data. To further validate the AlphaFold model, we tested the interaction between YonE and two SpbK[Ntd] mutants with double alanine substitutions (F69, F71 and F9, Y48) in gel-filtration assays (Fig. 7g). Both mutants disrupted the interaction with YonE, suggesting that both YonE clip domain:SpbK[NTD] and SpbK[NTD]:SpbK[NTD] interactions are required for stable complex formation. Together, these results demonstrate that the YonE clip domain serves as a platform for recruitment and oligomerisation of SpbK.

## Discussion

Our results provide a structural model for activation of the YonE-SpbK antiphage defence system in *Bacillus subtilis* (Fig. 8). SpbK-mediated antiphage defence begins when SpbK senses production of YonE portals upon SPβ phage infection[41,43]. In non-infected cells, SpbK exists as inactive dimers. The N-terminal domain is likely monomeric, but the TIR domains form symmetric dimers, presumably via the αC interface observed in the filament structure. As the active site of TIR NADases is found between two TIR domains in the BE interface, SpbK cannot cleave NAD$^+$ in this configuration. Upon SPβ-induced production of YonE portals, the N-terminal domains of SpbK dimers interact with YonE clip domains. The AlphaFold 3-predicted binding mode suggests that up to 6 SpbK dimers (12 N-terminal domains) can interact with one YonE portal. This high local SpbK concentration enables the TIR domains to self-associate via the BE interface observed in the filament and form up to 10 catalytically competent active sites per YonE portal.

This mechanism is consistent with Loyo and Grossman's recent study characterising the SpbK:YonE interaction in *B. subtilis* cells[43], and conceptually similar to recently described activation mechanisms for STING-TIR, SAVED-TIR, Pycsar antiphage defence systems[15,19,47,55], the programmed axon-death executioner SARM1[45] and plant immunity-associated NLRs (nucleotide-binding oligomerisation domain–like receptors)[50,51]. In all these systems, additional domains oligomerise and bring the TIR domains closer together, increasing their effective concentration, leading to active site formation and NADase activity[26].

The SpbK protofilaments conform to the general structure of TIR domain assemblies; that is, two open-ended strands of TIR domains arranged in either a parallel or antiparallel fashion[26]. Each strand in the SpbK protofilament features the evolutionarily conserved head-to-tail BE interface, which is required for signalling or NADase activity in all TIR systems characterised to date[15,27,29,30,32,45,47,50,51,56,57]. Like the STING-TIR filament structure[47], the SpbK protofilament has two antiparallel strands that associate via αC helix interactions. Most TIR domains exist as monomeric proteins in solution, but under conditions where their effective concentration increases, TIR monomers self-associate, forming two-stranded open-ended assemblies via intra- and inter-strand interactions. Interestingly, our data suggest that SpbK already exists as a stable dimer in solution via TIR αC helix interactions, suggesting that SpbK protofilaments are formed by self-association of TIR dimers.

The SpbK protofilaments self-associate, forming a hollow tube of five protofilaments. The TIR domains of the TLR signalling adaptors MAL and MyD88 also form higher-order assemblies consisting of multiple protofilaments: MAL oligomerises into a tube while MyD88 can either form a tube or a flat sheet[27,30,48,58]. For MAL and MyD88, only interactions within the protofilaments are of relevance to TLR signalling; the association of protofilaments into a tube or a sheet only occurs when these assemblies are reconstituted in solution, using a high concentration of purified protein. Given that active TIR NADase assemblies of SARM1, AbTir and STING-TIR only feature two strands[31,32,45,47], we speculate that only interactions within the SpbK protofilaments are required for NADase activity.

Central components in the phage replication cycle, including host takeover proteins, phage replication machinery and structural phage proteins, are commonly sensed by bacterial defence systems. Portal proteins assemble into dodecameric rings, which then interact with coat and scaffold proteins to form the procapsid. In the procapsid, the clip domains of the portal dodecamer serve as docking stations for both the terminase and tail machinery[59]. To facilitate TIR domain clustering, SpbK must interact with YonE after its assembly into a dodecameric ring, but before the YonE dodecamer forms a complex with the terminase or tail machinery. If phage portal dodecamers are produced in excess of terminases or tail proteins, free YonE dodecamers may bind SpbK and trigger NADase depletion.

Portal proteins are also sensed by some Avs antiphage defence systems[21]. Avs proteins are members of the STAND (signal transduction ATPases with numerous domains) family, which includes NLRs involved in plant and animal immunity. A cryoEM structure of EcAvs4 in complex with the PhiV-1 gp8 portal revealed that the TPR domain of EcAvs4 interacts with the stem, clip, and part of the wing domain of

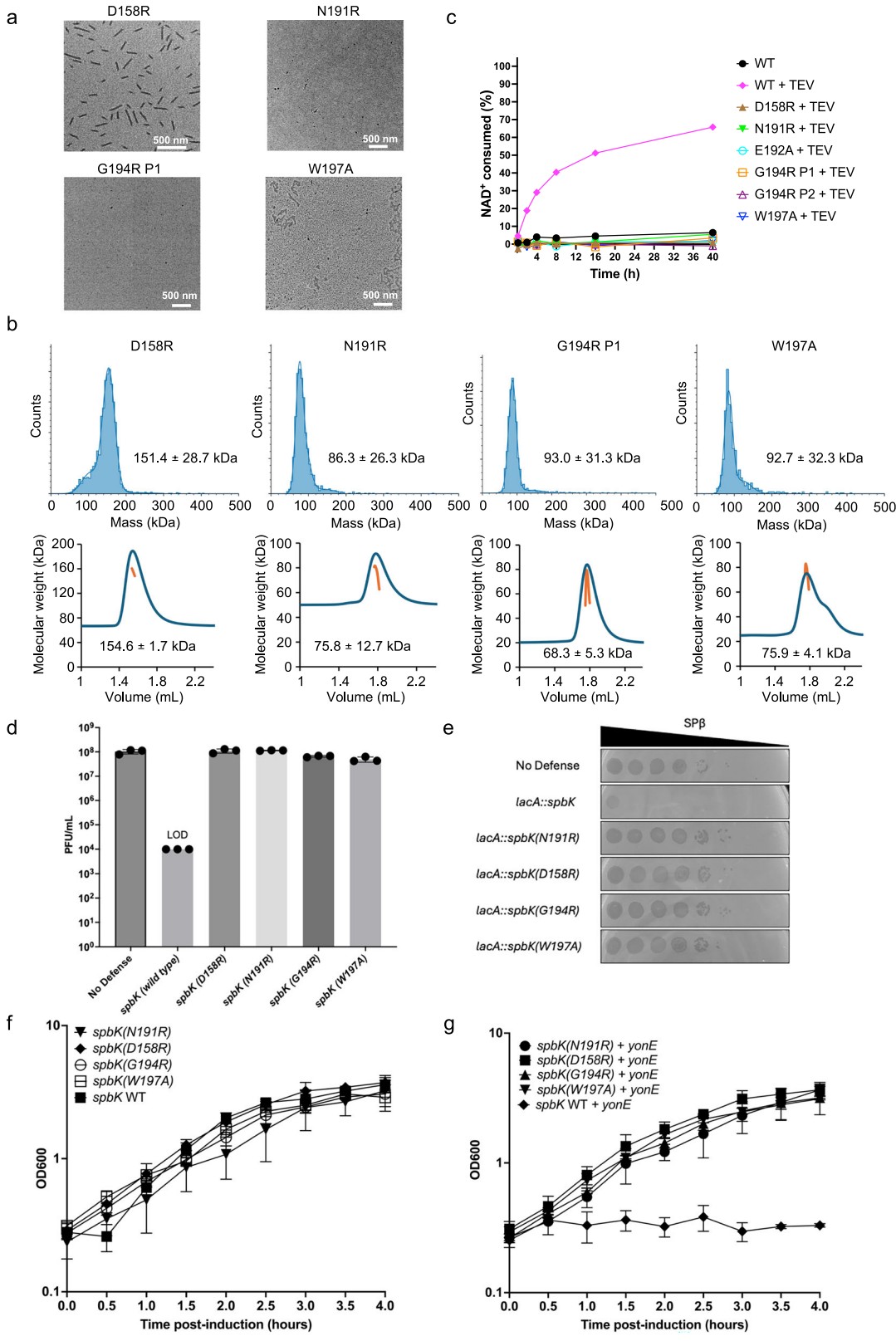

PhiV-1 gp8. However, the binding mode is not compatible with portal oligomerisation, suggesting that in this case, portal monomers are recognised by the defence system before they assemble into dodecamers.

Structural components of the phage, such as the portal, capsid, tail, and nucleic acids, are well-suited to facilitate TIR domain self-association to activate their NADase function. Roberts et al. recently

reported that the ThsB TIR domain proteins of a staphylococcal type 1 Thoeris defence system form complexes with the major capsid of phage Φ80α[60]. Wang et al. identified and characterised 12 TIR domain-containing antiphage defence systems in *E. coli* and found 25 phage genes that could trigger these systems. Many of these genes encoded proteins of diverse function, including ssDNA binding proteins, capsid proteins, and tail proteins[61]. Although not yet described

**Fig. 5 | Oligomerisation controls NADase activity and antiphage defence.**
**a** Representative negative-stain EM image of MBP-SpbK mutants incubated with TEV protease. For each mutant, micrographs were recorded from two grids of two independent samples. **b** Mass photometry and SEC-MALS analysis of MBP-SpbK mutants. Mass photometry experiments were conducted three times with similar results. In the SEC-MALS analysis the blue line represents the refractive index trace, while the orange line represents the average molecular mass distribution across the peak. **c** Reaction progress curves for MBP-SpbK mutants. The protein concentration was 0.2 μM, and the initial $NAD^+$ concentration was 500 μM. **d** Plaque-forming units (PFUs) per millilitre of SPβ plated onto cells with genotypes described on the bottom. All strains are in the CU1050 background, which enables larger plaque formation than other common lab strains. All four mutants of *spbK* display a phenotype similar to the no-defence control, indicating antiphage defence is abolished. LOD indicates limit of detection (no plaques detected at $1 \times 10^{-4}$ dilution). PFU/mL

values from three independent experiments are shown. Data are represented as mean ± SD. **e** Ten-fold serial dilutions of SPβ phage were spotted onto isogenic strains expressing no antiphage defence (CU1050, first row), *spbK* (CMJ534, 2nd row), or *spbK* mutants (CLL738-741, 3rd-7th rows). Large zones of clearing are indicative of a confluence of phage plaques and cell lysis. Small zones of clearing are indicative of individual or small clusters of phage plaques. **f** Growth curves of strains expressing various alleles of *spbK*. **g** Growth curves of strains expressing both *yonE* and various alleles of *spbK*. 1 mM IPTG was used to induce expression of *yonE*. IPTG was added directly after measuring OD600 at time 0. Cells expressing *yonE* and *spbK* mutants do not exhibit the growth arrest phenotype observed when *yonE* is expressed with wild-type *spbK*. *spbK* alleles are expressed under the native promoter of *spbK*. Data shown in **f**, **g** are from three biological replicates. Data are presented as mean ± SD. The wild-type *spbK* + *yonE* and wild-type *spbK* traces in (**f**, **g**) are from Loyo and Grossman, 2025[43].

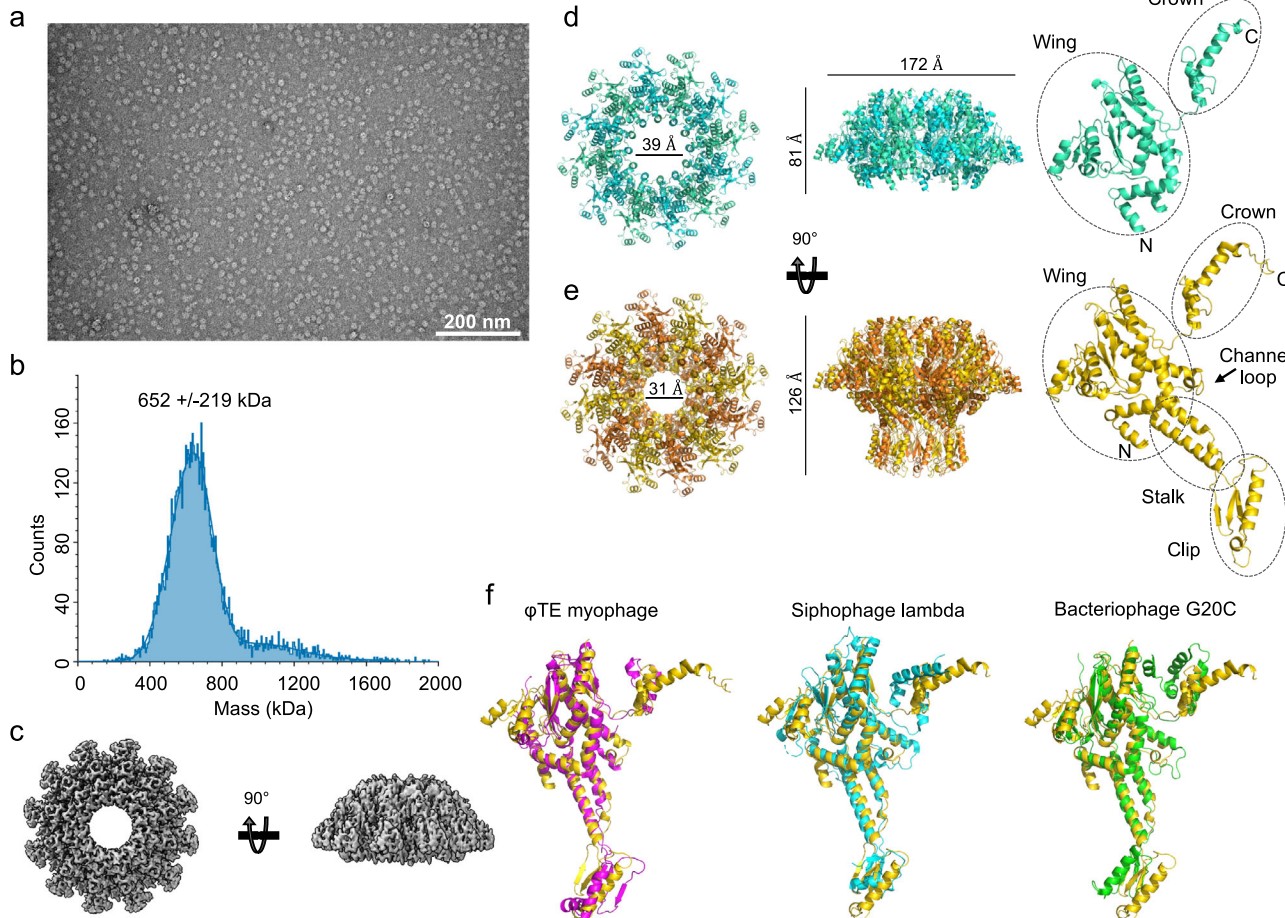

**Fig. 6 | YonE is a dodecameric portal protein. a** Representative negative-stain electron micrograph of YonE. Micrographs were recorded from three grids from three independent samples. **b** Mass photometry analysis of YonE with the calculated molecular weight. The theoretical molecular weight of a dodecamer is 665 kDa. **c** CryoEM density map of YonE dodecamer (top and side view). **d** Cartoon representation of the YonE cryoEM structure (top view, side view and individual monomer). Each sub-domain is highlighted by dashed circles. **e** Cartoon representation of the YonE AlphaFold 2 model (top view, side view and individual monomer). Each sub-domain is highlighted by dashed circles. **f** Structural superimpositions (Cα atoms) of YonE (gold) with the φTE myophage portal (PDB 9CC7; magenta; RMSD of 4.4 Å for 222 Cα atoms; 12% sequence identity), the siphophage lambda portal (PDB 8K38; cyan; RMSD of 3.9 Å for 216 Cα atoms; 6% sequence identity), and the bacteriophage G20C portal (PDB 4ZJN; green; RMSD of 3.8 Å for 225 Cα atoms; 6% sequence identity).

for any bacterial TIR system, some plant TIR domains with NADase activity have been shown to bind and self-associate on double-stranded DNA[62].

We discovered that SpbK produces N1-cADPR, a second messenger involved in calcium mobilisation[63,64], but its role in antiphage defence is not clear. The pro-neurodegenerative NADase SARM1 also produces N1-cADPR and has been implicated in promoting calcium

flux[33,65]. Interestingly, N1-cADPR was shown to induce expression of defence genes in tobacco[66]. SpbK is assumed to be an effector NADase, triggering abortive infection via rapid $NAD^+$ depletion[41,43]. However, co-expression of SpbK and YonE with the phage anti-defence system NARP1 ($NAD^+$ reconstitution pathway 1), which can reconstitute $NAD^+$ from ADPR and NAM, only partially suppresses the *B. subtilis* growth defect (Fig. S9). Although insufficiency of NARP1 to completely

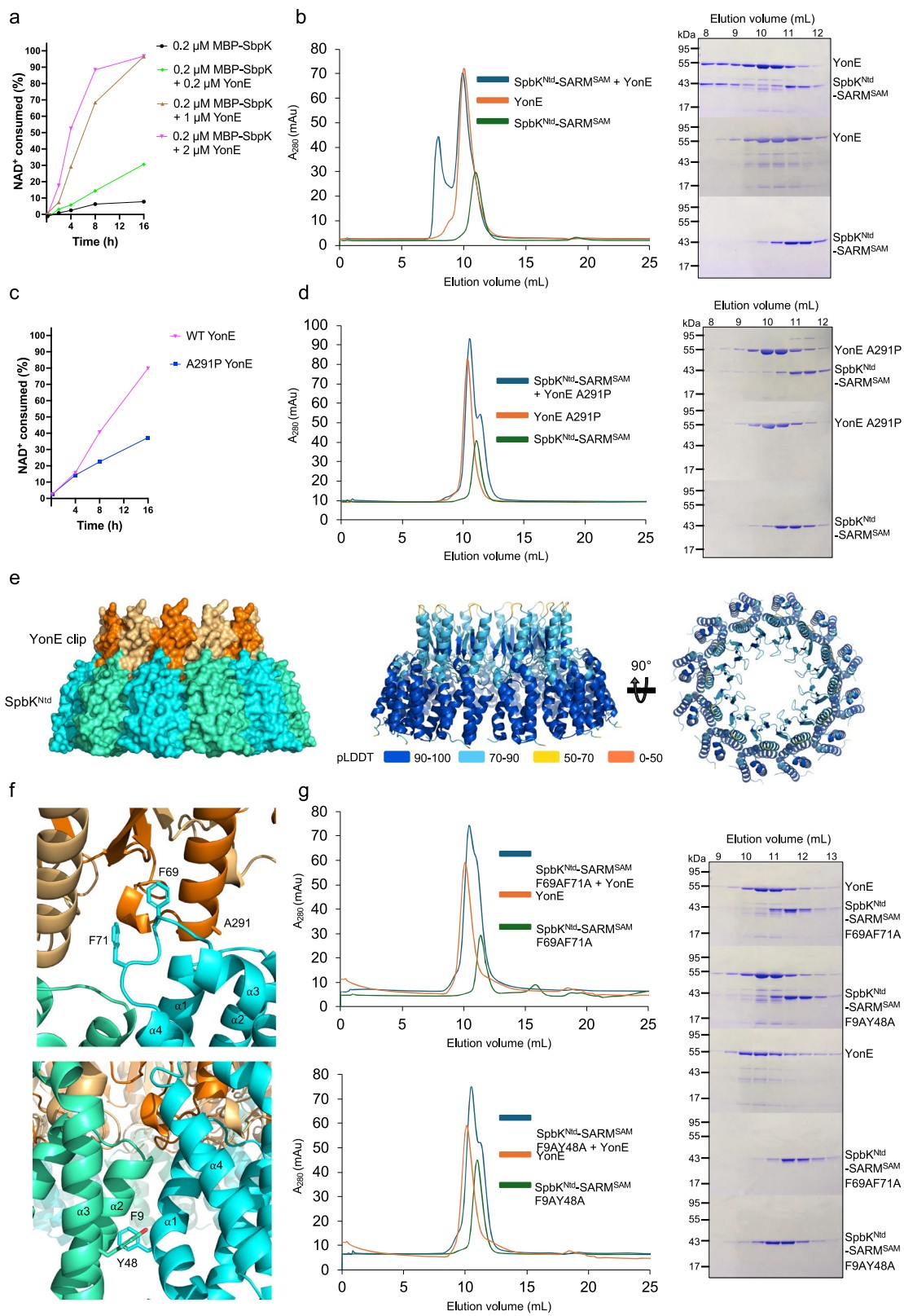

counteract NAD$^+$ depletion is a likely explanation for this partial phenotype, it is also possible that N1-cADPR plays a role in SpbK-mediated antiphage defence. Bacterial TIR NADases have been shown to trigger abortive infection via two distinct mechanisms: they can function as effectors themselves, rapidly depleting infected cells of NAD$^+$, or they can use NAD$^+$ as a substrate to produce immune signalling molecules such as 3′cADPR, N7-cADPR, and His-ADPR that activate effector proteins with distinct functions. 3′cADPR activates ThsA effectors of Thoeris type I systems. These effectors have SIR2 NADases that deplete infected cells of NAD$^+$ upon activation[18,32,38,67]. His-ADPR induces oligomerisation of transmembrane and macro domain (TM-macro)−containing ThsA effectors of type II Thoeris systems, resulting in membrane perturbation[36,37]. N7-cADPR is the most recently described signalling molecule produced by TIR domains. This molecule

**Fig. 7 | YonE activates the NADase function of SpbK. a** Reaction progress curves for 0.2 µM MBP-SpbK + YonE (concentration as labelled). The initial NAD$^+$ concentration was 500 µM. **b** Gel-filtration analysis of YonE:SpbK$^{Ntd}$-hSARM$^{SAM}$. Left: gel-filtration profiles. Right: SDS-PAGE analysis of peak fractions from SpbK$^{Ntd}$-SARM$^{SAM}$ + YonE (top), YonE (middle) and SpbK$^{Ntd}$-SARM$^{SAM}$ (bottom). **c** Reaction progress curves for 0.2 µM MBP-SpbK + 2 µM YonE A291P. The initial NAD$^+$ concentration was 500 µM. **d** Gel-filtration analysis of YonE A291P:SpbK$^{Ntd}$-hSARM$^{SAM}$. Left: gel-filtration profiles; Right: SDS-PAGE analysis of peak fractions from SpbK$^{Ntd}$-SARM$^{SAM}$ + YonE A291P (top), YonE A291P (middle) and SpbK$^{Ntd}$-SARM$^{SAM}$ (bottom). **e** AlphaFold 3 model of a dodecameric YonE-SpbK$^{Ntd}$ complex. Only the clip domain region of YonE (residues 270-225) was used for modelling. Left panel: surface

representation of the complex. YonE clip domain and SpbK$^{NTD}$ are highlighted in orange/light-orange and cyan/green-cyan, respectively. Right panel: YonE-SpbK$^{Ntd}$ complex coloured by the confidence metric, pLDDT (predicted local distance difference test). The pTM (predicted template modelling score) and ipTM (interface pTM) scores for the complex are 0.74 and 0.73, respectively. **f** Enlarged cutaway of the YonE-SpbK$^{Ntd}$ and SpbK$^{Ntd}$: SpbK$^{Ntd}$ interfaces in the dodecameric YonE-SpbK$^{Ntd}$ model. **g** Gel-filtration analysis of YonE:SpbK$^{Ntd}$-hSARM$^{SAM}$ F69AF71A and YonE:SpbK$^{Ntd}$-hSARM$^{SAM}$ F9AY48A. Left: gel-filtration profiles. Right: SDS-PAGE analysis of peak fractions. The gels in (**b**, **d**, and **g**) were stained with Coomassie brilliant blue. A280, absorbance at 280 nm; mAu, milli-absorbance units. The experiments in (**a**–**d**) and (**g**) were conducted twice with similar results.

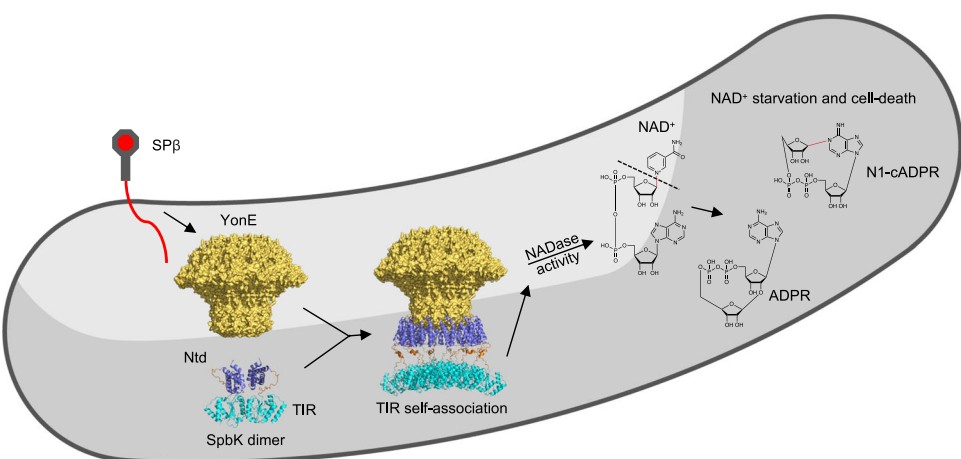

**Fig. 8 | Activation mechanism for the SpbK antiphage defence system.**

specifically activates a bacterial caspase-like protease of type IV Thoeris systems, which degrades cellular proteins to prevent phage replication[9]. In these Thoeris systems, the genes encoding for the TIR NADase and the nucleotide-binding effector are located in the same operon. Co-expression of *spbK* and *yonE* inhibited host-cell growth in the absence of any other ICE*Bs1* or SPß genes[41], indicating that if N1-cADPR is involved in antiphage defence, then the gene(s) encoding for N1-cADPR receptor(s) must be localised elsewhere in the *B. subtilis* genome. Further studies will be required to identify the *B. subtilis* targets of N1-cADPR and the role they play (if any) in SpbK-mediated antiphage defence.

## Methods
### Cloning
The cDNA of YonE (residues 46–506), MBP-SpbK, SpbK N-terminal domain (residues 1–94, SpbK$^{Ntd}$), SpbK$^{Ntd}$–SARM SAM domain (residues 409–548) fusion protein (SpbK$^{Ntd}$-SARM$^{SAM}$) and SpbK$^{Ntd}$–GST fusion protein (SpbK$^{Ntd}$-GST) were synthesised as gBlocks (Integrated DNA Technologies). Mutants of YonE (A291P), MBP-SpbK (D158R, N191R, E192A, G194R and W197A) and SpbK$^{Ntd}$-SARM$^{SAM}$ (F69AF71A and F9AY48A) were also synthesised as gBlocks (Integrated DNA Technologies). All primers used for cloning were obtained from Integrated DNA Technologies. YonE (wild-type and mutants), SpbK$^{Ntd}$, SpbK$^{Ntd}$-SARM$^{SAM}$ (wild-type and mutants) and SpbK$^{Ntd}$-GST were amplified by PCR and cloned into the pMCSG7 vector using ligation-independent cloning[68]. MBP-SpbK (wild-type and mutants) was cloned into the pET28b vector using the Gibson Assembly reaction[69]. Pure plasmids were prepared using the QIAprep Spin Miniprep Kit (Qiagen), and the sequences were confirmed by the Australian Genome Research Facility.

### Protein production and purification
Wild-type and mutant SpbK and YonE proteins (Table S4) were produced in *E. coli* BL21 (DE3) cells, using the autoinduction method[70].

They were purified to homogeneity, using a combination of immobilised metal-ion affinity chromatography (IMAC) and size-exclusion chromatography (SEC). The cells were grown at 37 °C, until an OD600 of 0.6–0.8 was reached. The temperature was then reduced to 20 °C, and the cells were grown overnight for approximately 16 h. The cells were harvested by centrifugation at 5000 × *g* at 4 °C for 15 min, and the cell pellets were resuspended in 2–3 mL of lysis buffer (50 mM HEPES pH 8.0, 500 mM NaCl) per g of cells. The resuspended cells were lysed using a digital sonicator and clarified by centrifugation (15,000 × *g* for 30 minutes). The clarified lysate was supplemented with imidazole (final concentration of 30 mM) and then applied to a nickel HisTrap column (Cytiva) pre-equilibrated with 10 CVs of wash buffer (50 mM HEPES pH 8.0, 500 mM NaCl, 30 mM imidazole) at a rate of 4 mL/min. The column was washed with 10 CVs of wash buffer, followed by elution of bound proteins using elution buffer (50 mM HEPES, pH 8, 500 mM NaCl, 250 mM imidazole). The elution fractions were analysed by SDS-PAGE, and the fractions containing the protein of interest were pooled and further purified on either a S75 HiLoad 26/600 column (SpbK$^{Ntd}$) or a S200 HiLoad 26/600 column (YonE, MBP-SpbK, SpbK$^{Ntd}$-SARM$^{SAM}$ and SpbK$^{Ntd}$-GST) pre-equilibrated with gel-filtration buffer (10 mM HEPES pH 7.5, 150 mM NaCl). The peak fractions were analysed by SDS-PAGE, and the fractions containing the protein of interest were pooled, concentrated (SpbK$^{Ntd}$, 31 mg/mL; wild-type and mutant YonE, 29-42 mg/mL; wild-type and mutant MBP-SpbK, 2–10 mg/mL; wild-type and mutant SpbK$^{Ntd}$-SARM$^{SAM}$, 40-59 mg/mL; SpbK$^{Ntd}$-GST, 39 mg/mL), flash-frozen as 10 µL aliquots in liquid nitrogen, and stored at −80 °C.

### NMR-based enzymatic assays
NMR samples were prepared in 175 µL HBS buffer (50 mM HEPES, pH 7.5, 150 mM NaCl), 20 µL D$_2$O, and 5 µL DMSO-d6, resulting in a total volume of 200 µL. Each sample was subsequently transferred to a 3 mm Bruker NMR tube. All $^1$H NMR spectra were acquired with either a

Bruker Avance Neo 600 MHz NMR spectrometer equipped with a QCI-F cryoprobe, a Bruker Avance III HD 800 MHz NMR spectrometer equipped with a TCI cryoprobe, or a Bruker Avance III HD 600 MHz NMR spectrometer equipped with a TCI cryoprobe at 298 K. To suppress resonance from $H_2O$, a water suppression pulse programme (P3919GP) using a 3-9-19 pulse-sequence with gradients[71] was implemented to acquire spectra with an acquisition delay of 2 s and 32 scans per sample. For each reaction, spectra were recorded at multiple time points such as 10 min, 2 h, 4 h, 8 h, and 16 h, depending on instrument availability. All spectra were processed by TopSpin™ 4.4.0 (Bruker) and MNova 14.3.0 (Mestrelab Research). The reaction progression was calculated based on the integration of non-overlapping resonance peaks, which vary depending on sample composition, from pairs of reactant and product, such as $NAD^+$ and NAM, respectively. The detection limit (signal-to-noise ratio >2) was estimated to be 10 μM.

### Production and purification of N1-cADPR from SpbK

To conclusively verify the identity of SpbK-produced N1-cADPR, production reactions for N1-cADPR were performed using conditions similar to the $^1H$ NMR NADase assay. A solvent volume of 4 mL was used for each reaction, consisting of HBS buffer (50 mM HEPES, pH 7.5, 150 mM NaCl) with 0.5 μM MBP-SpbK and 5 mM $NAD^+$. These reactions were performed at room temperature and monitored intermittently by $^1H$ NMR. To stop the reactions, the $His_6$-tagged protein was removed by incubating the mixture with 200 μl of HisPur nickel–nitrilotriacetic acid resin for 30–60 min. The resin was subsequently removed by centrifugation at 1500 x g for 5 min, and the supernatant was subjected to HPLC-based separation to purify the base-exchange products. A Shimadzu Prominence HPLC equipped with a Synergi 4-μm Hydro-RP 80-Å column was used for separation. The mobile phase consisted of phase A (0.04% v/v TFA in water) and phase B (0.04% v/v TFA in acetonitrile). Different gradients, flow rates, and run times were applied depending on prior optimisation with individual reaction mixtures. Product peaks were confirmed by comparison with individual chromatograms of $NAD^+$, NAM, and ADPR. Fractions corresponding to the N1-cADPR product peaks were collected, concentrated, lyophilised and stored at −20 °C.

NMR spectroscopy and high-resolution mass spectrometry (HRMS) were employed to characterise N1-cADPR produced by SpbK and compare it with a commercial sample of N1-cADPR purchased from Merck (C7344) for structural validation. About 2 mg of purified N1-cADPR was dissolved in 200 μL of $D_2O$. The sample was transferred to a 3 mm NMR tube. The Bruker Avance Neo 600 MHz NMR spectrometer was utilised to acquire $^1H$ (Fig. S2a), $^{13}C$, $^1H$-$^1H$ COSY, $^1H$-$^{13}C$ HSQC, and $^1H$-$^{13}C$ HMBC spectra at 298 K, which informed the assignments of $^1H$ and $^{13}C$ peaks and correlations (Table S1), especially those linking the adenine ring and the two ribose rings (Fig. S2c). HRMS showed SpbK-produced N1-cADPR has the same m/z ratio and a similar fragmentation pattern to the commercial sample (Fig. S2b).

### Preparation of phage-infected cell lysate for LC-HRMS analysis

Lysates were prepared as described previously[46], with minor modifications described below. An overnight culture of *B. subtilis* CMJ82 was diluted 1:50 into 200 mL fresh LB. The diluted cultures were incubated at 37 °C, 220 rpm for 3 – 4 hours until $OD_{600nm}$ ~ 0.6. 40 mL of the culture was removed as a $t = 0$ min sample and immediately centrifuged at 10,000 × g, 4 °C for 7 min. The supernatant was discarded, and the cell pellet was temporarily stored at −80 °C. Then, 200 mL of SPBeta clear-plaque phage (~ 7 × $10^9$ PFUs/mL) was added to the host cells to reach MOI > 10. The infected cell culture was incubated at 30 °C, 220 rpm, and 40 mL of the culture was removed every 10 min to be immediately centrifuged at 10,000 × g, 4 °C for 7 min. The supernatant was discarded, and the cell pellet was temporarily stored at −80 °C. The cell pellets were thawed at room temperature and resuspended in 600 μL of 100 mM sodium phosphate buffer (pH = 7) +

4 mg/mL lysozyme. After incubation at room temperature for 10 min, the cells were transferred into 2 mL tubes with Lysing Matrix B (MP Biomedicals #116911050) and lysed using an Omni Bead Ruptor 12 for 2 × 40 s at 6 m/s with a dwell time of 4 min in between. After lysis, the tubes were centrifuged at 20,000 × g, 4 °C for 5 min. Then, 500 μL of each supernatant was transferred to Amicon Ultra-0.5 Centrifugal Filter Units 3 kDa (EMD Millipore #UFC500396) and centrifuged for 20 min at 13,500 × g, 4 °C. The filtrate was collected, and 10 μL of each was used for LC-MS analysis.

### Preparation of IPTG-induced cell lysate for LC-HRMS analysis

Lysates were prepared as described previously[46] with minor modifications described below. Overnight culture of *B. subtilis* CMJ685 was diluted 1:100 into 400 mL fresh LB. The diluted cultures were incubated at 37 °C, 220 rpm for 3 - 4 hours until $OD_{600nm}$ ~ 0.3. 40 mL of the culture was removed as a $t = 0$ min sample and immediately centrifuged at 10,000 × g, 4 °C for 7 min. The supernatant was discarded, and the cell pellet was temporarily stored at −80 °C. Then, 3.6 mL 100 mM IPTG was added to the host cells to reach a final concentration of 1 mM, inducing the expression of YonE. The induced cell culture was incubated at 30 °C, 220 rpm, and 40 mL of the culture was removed every 15 min to be immediately centrifuged at 10,000 × g, 4 °C for 7 min. The supernatant was discarded, and the cell pellet was temporarily stored at −80 °C. The cell pellets were thawed at room temperature and resuspended in 600 μL of 100 mM sodium phosphate buffer (pH = 7) + 4 mg/mL lysozyme. After incubation at room temperature for 10 min, the cells were transferred into 2 mL tubes with Lysing Matrix B (MP Biomedicals #116911050) and lysed using an Omni Bead Ruptor 12 for 2 × 40 s at 6 m/s with a dwell time of 4 min in-between. After lysis, the tubes were centrifuged at 20,000 × g, 4 °C for 5 min. Then, 500 μL of each supernatant was transferred to Amicon Ultra-0.5 Centrifugal Filter Units 3 kDa (EMD Millipore #UFC500396) and centrifuged for 20 min at 13,500 × g, 4 °C. The filtrate was collected, and 10 μL of each was used for LC-MS analysis.

### LC-MS/MS analysis of N1-cADPR in the phage-infected cell lysate & IPTG-induced-cell lysate

Liquid chromatography analysis was performed on an ACQUITY UPLC I-Class PLUS System using a Luna Omega 5-μm Polar C18 100 Å column (250 × 4.6 mm). The mobile phase A was water + 0.1 % (v/v) formic acid, and the mobile phase B was acetonitrile + 0.1 % (v/v) formic acid. The flow rate was kept at 0.7 mL min⁻¹ and the gradient was as follows: 0% B (0–10 min), increase to 2.5% B (10–15 min), increase to 5% B (15–16 min), increase to 95% B (16–17 min), hold 95% B (17–27 min), decrease to 0% B (27–28 min), hold 0% B (28–38 min). High-resolution electrospray ionisation (HR-ESI) mass spectra with collision-induced dissociation (CID) MS/MS were obtained using a Waters Synapt G2S Quadrupole Time-of-Flight (QTOF). The instrument was operated in negative ionisation mode. The MS spectra were obtained on the Time-of-Flight analyser with a scan range of 300–800 Da and analysed using MassLynx 4.1 software. The m/z of interest was filtered through the quadrupole, subjected to CID (energy ramp 34–44 V), and analysed on the Time-of-Flight analyser with a scan range of 50–750 Da.

### Pre-purification of IPTG-induced-cell lysate

The liquid chromatography pre-purification was performed on an Agilent 1260 Infinity II HPLC system using a Luna Omega 5 μm Polar C18 100 Å column (250 × 4.6 mm). The mobile phase A was water +0.1 % (v/v) formic acid, and the mobile phase B was acetonitrile + 0.1 % (v/v) formic acid. The flow rate was kept at 0.7 mL min⁻¹ and the gradient was as follows: 0% B (0–15 min), increase to 100% B (15–16 min), hold 100% B (16–26 min), decrease to 0% B (26–27 min), hold 0% B (27–37 min). 9–12 min fractions were collected, dried under lyophilisation and redissolved with 100 μL water for further MS analysis.

## Analytical size-exclusion chromatography

After incubation, 500 μL of each sample was loaded onto a S200 10/300 column (Cytiva) using an Akta Pure device (Cytiva) at 0.5 mL/min. Analysis of chromatograms and image preparation were performed in Unicorn 7 software (Cytiva) and Excel (Microsoft). Peak fractions were analysed by SDS-PAGE.

## Mass photometry

Mass photometry data were collected on a Refeyn OneMP system (Refeyn Ltd, Oxford, UK). High precision glass coverslips (Marienfeld Ltd) were thoroughly washed three times using Milli-Q® H₂O, ethanol and 2-propanol, followed by drying under a stream of nitrogen gas. For sample loading, a silicon gasket (Refeyn Ltd) was placed on the slide. To focus the objective, 16 μL of buffer (10 mM HEPES pH 7.5, 150 mM NaCl) was placed in a well. Thereafter, 4 μL of the test sample was loaded and mixed by pipetting the volume up–down five times, followed by immediate measurement. Mass photometry data were acquired and analysed using AcquireMP and DiscoverMP software (Refeyn Ltd), respectively. A Gaussian curve was fitted to the resulting peaks in the contrast histogram (depicting the number of detected molecules and their corresponding contrast value), where each peak represented a subpopulation of molecules with a particular molecular weight (MW). The contrast values of observed counts were converted into MW by calibration using a control protein of known MW and size; in-house bovine serum albumin (BSA), with a monomer at 66 kDa, a dimer at 122 kDa, and a trimer at 198 kDa.

## Size-exclusion chromatography–coupled multiangle light scattering

A DAWN HELEOS II 10-angle light scattering detector coupled with an Optilab rEX refractive index detector (Wyatt Technology) combined with a Superdex 200 5/150 Increase size-exclusion column (Cytiva) and connected to a Prominence HPLC (Shimadzu), was used for SEC-MALS. The column was equilibrated in gel-filtration buffer, and 30 μL of the purified protein was run through the column at 0.25 mL/min. Molecular masses were calculated using Astra 6.1 (Wyatt Technology).

## Negative-stain EM

4 μL of sample was placed on a carbon-coated copper grid and incubated for 60 s. The grid was then washed with Milli-Q® H₂O and stained with 1% uranyl acetate for 60 s and air-dried. The images were collected on a JEOL JEM-1400Flash TEM transmission EM at ×20,000-60,000 magnification at 80 keV.

## CryoEM data collection

YonE: Purified YonE (0.75 mg/mL; 12 μM) was loaded onto a glow-discharged Quantifoil grid (R1.2/1.3 300-mesh holey carbon film coated, ProSciTech), blotted for 5.5 s under 95% humidity at 4 °C, and plunged into liquid ethane, using a FEI Vitrobot Mark 2 automatic plunge freezer (Thermo Fisher Scientific). For data collection, movies (0° and 30° tilt) were acquired on a JEM-Z300FSC (Cryo-ARM300, JEOL Ltd.) operating at an acceleration voltage of 300 keV equipped with an in-column Omega energy filter set with a slit opening of 20 eV. Movies were recorded using a K3 Summit direct electron detector (Gatan) at a magnification of ×100,000 corresponding to a pixel size 0.48 Å at the specimen level. All movies were exposed using a total dose of 40 e⁻/Å² over 40 frames and a defocus range between −0.5 and −2.5 μm.

SpbK: Filaments were obtained by incubating purified MBP-SpbK (1 mg/mL; 10 μM) with TEV protease (5 μM) in HBS buffer supplemented with 10% glycerol for 1 h at 25 °C. The filament sample was then loaded onto a glow-discharged Quantifoil grid (R2/1 300-mesh holey carbon film coated, ProSciTech), blotted for 8.5 s under 95% humidity at 4 °C, and plunged into liquid ethane, using a Leica EMGP2 Automatic Plunge Freezer (Leica Microsystems). For data collection, movies were acquired on a JEM-Z300FSC (Cryo-ARM300, JEOL Ltd.) operating at 300 keV equipped with an in-column Omega energy filter. Movies were recorded with a K3 Summit direct electron detector (Gatan) operating at ×100,000 magnification (0.48 Å per pixel). All movies were exposed using a total dose of 40 e⁻/Å² over 40 frames and a defocus range between −0.5 and −2.5 μm.

## CryoEM data processing

All processing steps were performed using CryoSPARC v4.5.1 and v4.6.2[72], and the cryoEM processing workflows are summarised in Figs. S3 and S5.

YonE: A total of 4089 movies were collected and imported into CryoSPARC. Alignment of movie frames was performed using patch-based motion correction. Fitting of the contrast transfer function and defocus estimation was performed using patch-based CTF estimation. Particle picking was performed using the Blob picker in CryoSPARC, and 246,156 particles were extracted with a 700 px box size (Fourier cropped to 256 px). Several rounds of 2D classification were performed to remove inferior particles. 22,320 particles representing the best 2D classes were then used for ab-initio reconstruction using C1 symmetry, followed by homogeneous and non-uniform refinement with C12 symmetry imposed. The final resolution of the 3D reconstruction was 3.3 Å, based on the gold standard FSC 0.143 criteria. Local resolution distribution was evaluated using CryoSPARC. The statistics are listed in Table S3.

SpbK: A total of 11,993 movies were collected and imported into CryoSPARC. Alignment of movie frames was performed using patch-based motion correction. Fitting of the contrast transfer function and defocus estimation was performed using patch-based CTF estimation. Filament segments were auto-picked by "Filament Tracer" using a filament diameter of 100 Å and a separation distance of 50 Å. The resulting filament segments were extracted with an 800 px box size (Fourier cropped to 400 px). Iterative rounds of 2D classification were performed to remove junk classes and poor-quality filaments. 3D reconstructions of the final particle set containing 116,455 filament segments were performed using Helical Refinement. A reconstruction with recognisable protein features (α-helices, β-strands and side chain densities) was obtained using C1 symmetry, a helical rise of 8.9 Å and twist of 79.3° per subunit. Inspection of this reconstruction revealed that the filament contains 5 protofilaments, each consisting of two antiparallel TIR domain strands with dihedral (D1) symmetry. The quality of the map was further improved by applying D1 symmetry and non-uniform regularisation to the Helical Refinement protocol. The final resolution of the 3D reconstruction was 3.3 Å, based on the gold standard FSC 0.143 criteria. The final volume was sharpened with local resolution filtering automatically estimated in CryoSPARC. The statistics are listed in Table S3.

## Model building and refinement

YonE: ModelAngelo[73] supplied with the YonE sequence was used to build an initial model of the entire dodecamer into the map. An AlphaFold 2 model of a YonE monomer was subsequently fitted into the map by structural alignment to the ModelAngelo model. The fitted model was subjected to iterative rounds of model building and refinement using Coot[74] and phenix.real_space_refine from the PHENIX suite[75,76]. The full model was generated by applying symmetry operators and refined further using Coot and phenix.real_space_refine.

SpbK: ModelAngelo[74] supplied with the SpbK sequence was used to build an initial model into the helical refinement map. To capture all protein-protein interfaces in the filament, 10 chains from the ModelAngelo model were subjected to iterative rounds of model building and refinement using Coot[74] and phenix.real_space_refine from the PHENIX suite[75,76].

## Structure prediction of YonE and YonE:SpbK complex

YonE: A model of dodecameric YonE was generated using AlphaFold 2 Multimer[52,77] implemented in the ColabFold interface (v1.5.1), using a local ColabFold installation with default settings.

YonE:SpbK complex: AlphaFold 3[78] (https://AlphaFoldserver.com) was used to generate a model of dodecameric YonE in complex with 12 copies of SpbK$^{Ntd}$. Only the clip domain region (residues 270-325) of YonE was used for modelling. The YonE clip domain:SpbK$^{Ntd}$ and SpbK$^{Ntd}$:SpbK$^{Ntd}$ interfaces had ipTM scores of 0.56–0.60 and 0.71–0.73, respectively (Fig. S8). To accommodate modelling complexes of full-length SpbK and near full-length YonE (residue 46–506), we ran a local implementation of AlphaFold 3 (commit: 69b749c) with pair_attention_chunk_size set to 8, pair_transition_shard_spec set to 8 and unified memory enabled. Inference was performed using an Nvidia H200 GPU (141 GB VRAM). To sample candidate predictions, we produced 10 independent replicates with 5 diffusion samples per seed. One of the predicted models displayed similar interfaces to both the truncated YonE:SpbK$^{Ntd}$ model and the SpbK TIR domain filament, but the chain ipTM scores were very low ( ~ 0.15), and the model has only been used for illustrative purposes in Fig. 8.

## Testing of SpbK mutants in B. subtilis

Media and growth conditions: *B. subtilis* cells were grown in LB or S7$_{50}$ minimal medium supplemented with 0.1% glutamate and 1% glucose as a carbon source. Isopropyl β-D-1-thiogalactopyranoside (IPTG) was used at 1 mM to induce expression from the LacI-repressible IPTG-inducible promoter Pspank(hy). Antibiotics used for selection in *B. subtilis* included: kanamycin (5 μg/mL), tetracycline (10 μg/mL) and spectinomycin (100 μg/mL).

For typical experiments, a culture was started from a single colony (after overnight growth) and grown at 37 °C to mid-late exponential phase in the indicated medium. Cells were diluted into fresh medium for continued growth as indicated. For experiments with genes expressed from an inducible promoter, cultures were split at the time of dilution into medium with and without inducer, as indicated.

Temperate phages were induced by the addition of mitomycin C (MMC) to 1 μg/mL to a lysogen growing exponentially in LB medium at 37 °C. After MMC induction, cells were grown for one additional hour and pelleted at 4000 × *g* for 5 minutes in a tabletop centrifuge. The lysate was transferred to a new tube, 1:100 v/v chloroform was added to inhibit cell growth, and stored at 4 °C.

*Efficiency of plating assays:* Phage stocks were diluted in phage buffer (150 mM NaCl, 40 mM Tris-Cl, 10 mM MgSO$_4$) and mixed with 300 μL of cells that were growing exponentially in LB medium. Cells and phage were incubated for 5 minutes at room temperature to allow adsorption, transferred to 3 mL of molten LB + 0.5% Bacto-agar, and plated onto warm LB agar plates. Plates were incubated at 30 °C overnight to allow plaque formation. Serial dilutions (typically 10-fold) for spot assays were made in phage buffer, and 2 μL were spotted onto a lawn of cells on LB agar plates. Phage plaquing efficiencies were determined essentially as described[41]. Briefly, 100 μL of various dilutions of a phage stock were mixed with 300 μL of cells that had been grown to OD600 of ~0.5 (~3 × 10$^7$ cells), incubated at room temperature for 5 minutes, mixed with 3 mL LB top agar and spread onto an LB agar plate. Plaques were counted after incubation overnight at 30 °C and are presented as plaque-forming units per mL (PFU/mL) of phage stock.

*Strains and alleles:* *B. subtilis* strains used in this study are listed in Table S5. All strains are derivatives of PY79[79] or CU1050[80], both of which are cured of SPβ and ICE*Bs1*. Indicated alleles were introduced by natural transformation[81] and appropriate selection. The *spbK* mutant alleles were constructed by PCR and Gibson isothermal assembly with primer overhangs encoding the desired mutation. Genomic DNA used for construction was derived from CMJ684 (PY79,

*lacA*::{*spbK kan*}). *B. subtilis* cells were transformed via natural transformation, with Gibson assembly reaction added to competent cells.

## Reporting summary

Further information on research design is available in the Nature Portfolio Reporting Summary linked to this article.

## Data availability

The cryoEM density maps have been deposited in the Electron Microscopy Data Bank (EMDB) under accession numbers EMD-71643 (YonE dodecamer) and EMD-71644 [https://www.ebi.ac.uk/pdbe/entry/emdb/EMD-71644] (SpbK filament). The atomic coordinates have been deposited in the Protein Data Bank (PDB) under accession numbers 9PHA (YonE dodecamer) and 9PHB (SpbK filament). Protein structures used for analysis in this study are available in the Protein Data Bank under accession codes 4ZJN, 7NAK, 7UN8, 7UXU, 8K38, 8SPO and 9CC7. The source data underlying Figs. 1c, 3a–c, 3e, 5b–g, 6b, 7a–d, 7g and Supplementary Figs. 1c, 4b, 7a–c, and 9b are provided as a Source Data File. Source data are provided with this paper.

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

## Acknowledgements
We acknowledge the Centre for Microscopy and Microanalysis, University of Queensland and staff and the Mark Wainwright Analytical Centre Structural Biology Facility, University of New South Wales. The work was supported by the National Health and Medical Research Council (Investigator Grant 1196590 to T.V., Investigator Grant 2025931 to B.K.) and the Australian Research Council (Future Fellowship (FT200100572) to T.V. and Discovery Early Career Research Award (DE250101258) to Y.S.).

## Author contributions
Writing—original draft and conceptualisation, B.M., Y.S., T.V.; investigation, B.M., Y.S., C.L., Y.C., T.L., G.M., L.B., T.V., T.M., V.M., P.R., W.G.; writing—review & editing, all authors; resources, T.V.; methodology, B.M., Y.S., C.L., T.L., Y.C., B.L., T.V., T.M., V.M., W.G.; data curation, Y.S., T.M.; validation, B.M., Y.S., C.L., T.L., Y.C., L.B., T.V., B.M., T.M., V.M., W.G.; funding acquisition, T.V.; supervision, T.V., S.R., Y.S., A.G., J.G., B.K., B.L.; formal analysis, Y.S., T.M., B.L.; visualisation, B.M., Y.S., C.L., T.V.; project administration, T.V.

## Competing interests
The authors declare no competing interests.
