## [Transparent Peer Review file · Nature Communications]

Molecular characterisation of the *Bacillus subtilis* SpbK antiphage defence system

Corresponding Author: Dr Thomas Ve

Version 0:

Reviewer comments:

Reviewer #1

(Remarks to the Author)

In this work, the authors have demonstrated YonE is a dodecameric portal protein that activates the NADase function of SpbK by facilitating TIR domain clustering. These results may provide insight into how bacterial TIR NADases recognise phage infection. While the authors have performed detailed structural determination of proteins yet certain aspects remain unclear.

Limitations

There are two major criticism that this reviewer had regarding the manuscript. The mechanistic details of the enzymatic reaction remains unclear despite the detailed cryoEM structure determination and the series of mutants used in this work. First, TIR domain is essential for enzymatic reaction of SPBK, which is clearly demonstrated in this work. However, the experimental design of the work is convoluted by polymerization of the TEV cleaved MBP-SPBK. While the MW of MBP-SPBK wild type and mutants D158R, N191R, E192A, W197A etc are similar (Figure 5), yet the enzymatic activity is present only for the TEV-cleaved WT. Does it imply that TEV-cleaved Mutants donot oligomerize ? Or does it oligomerize in a manner that is not conformationally competent for the reaction. This reviewer strongly suggest (a) Structural studies of NAD⁺ (non hydrolyzable analogue) or (b) mutant protein that has intermediate activity or (c) constitutively active monomer/dimer. This is especially pertinent since the Figure 3 demonstrates clearly that MBP-SPBK and TEV cleaved MBP_SPBK has drastically different kinetics. While oligomerization could be essential , Is it possible that the mode of polymerization influence the kinetics of the reaction? Second, the impact of YonE on the reaction mechanism of SPBK remains unclear to this reviewer. While the graph in The SEC profiles suggest that the two proteins interact, yet the relevance of this interaction in the NAD⁺ consumption is unclear to this reviewer. The graph is figure 7a and 3e suggests that kinetics of the control SPBK is different in two experiments atleast in 16 hrs. Unless the mechanistic details of the reaction mechanism is clarified , the utility of Fig 8 remains inconsistent with the claims of this paper.

Minor point

While NMR has been used to monitor the enzymatic reaction of NAD consumption, an orthogonal method might be required, since there can be contribution from NH signals of the protein side chain.

Reviewer #2

(Remarks to the Author)

The work in this paper is significant because it illustrates yet another mechanism in the arsenal of bacterial defense systems. The authors successfully demonstrate that TIR-containing SpbK, when in the presence of the phage particle YonE, increases the NADase activity required to defend against pathenogenic activity.

The structural mechanism proposed could benefit from a few more experimental ideas: could cryoEM be performed on the stable dimer form of SpbK, and also, SpbK in complex with YonE? Both experiments could further illustrate the mechanistic

story and may be simple to do since purification was already performed of each. Also, for the cryoEM reconstruction of YonE, I am curious to see the full computational processing workflow. Was any symmetry or 3D classification performed prior to the final reconstruction of this 12-mer density of YonE? If so, it would be ideal to inspect the C1 solution prior to any imposed symmetry, as well as any other resulting 3D classes, to see if the missing base units can be found. This could be included in the supplemental processing figure, but it would lend to a better understanding or interpretation of the missing density, which can potentially get averaged out if it does not follow an imposed symmetry rule.

Overall the findings do support and elucidate SpbKs mechanism in the fight against SpB phage infection, and it adds to the expansive set of tools available to bacterial immune systems.

Reviewer #3

(Remarks to the Author)

SpbK is a TIR domain containing protein from Bacillus, shown previously to be an enzyme with NAD⁺ depletion activity which presumably leads to cellular growth arrest in response to phage infection. The authors have elucidated the structure of higher order filaments using cryo-EM and established structural similarity of these filaments to other known TIR-domain containing proteins. In addition, the authors describe interactions between YonE (a portal protein from phage SPbeta) and SpbK. Based on separate structural determinations of SpbK and YonE complemented with AlphaFold predictions and mutagenesis experiments, the authors have determined residues/interfaces (namely, SpbK's N terminal domain to YonE's clip domain) which are key to interactions between SpbK and YonE.

The authors have presented us with a very thorough study that has layers of details which will of interest to biochemists, structural biologists, phage biologists, and those studying immune signaling. This represents one of the few research articles in recent memory that are explicit about the pitfalls of studying cyclic ADPR analogs and that provide clear and in-depth analysis of these molecules. The story is clear and the data and analysis are original and exciting. Minor comments mainly on the formatting of figures and suggestions for one or two additional experiments to solidify alphafold predictions can be found below.

- First results subheading “an” NADase not “a”
- For the double-peaks observed for N1-cADPR: Can the authors speculate more on what component in lysate could be causing this phenomenon? Metals perhaps? Could the second peak which disappears following pre-purification be a less-stable isomer which simply does not survive the lyophilization procedure?
- When discussing alphafold prediction of the hexamer bound to NAD⁺, the authors should consider adding mutagenesis data to supplement the supposition. Mutants of residues suspected to form the composite active site are especially interesting (perhaps K224 stands out as the authors mention this residue specifically).
- Can the authors please comment here and in the main text why they chose to remove the first 45 amino acids of the YonE construct used for cryo-EM analysis? The statement that the residues were truncated is made but no justification is provided.
- In experiments mixing YonE and SpbK WT/mutants, why were those concentrations tested? (2uM and 0.2 uM, respectively)? If this is simply due to the expected stoichiometry of the final complex, then the authors should amend the manuscript with this detail for completeness and/or conduct additional experiment - titration of [YonE] into SpbK.
- Can the authors share the SpbK NTD to YonE monomer to monomer alphafold prediction (possibly as a supplementary figure)? The statistics might look better and still provide useful information regarding the binding interactions.
- Figures with SEC-MALS should clearly describe in the figure legends what is indicated by the orange trace observed in the graphs.
- Figure 5f and 5g- The wildtype spbK + yonE trace needs to be presented as well for appropriate comparison and indeed it might be beneficial to combine both panels into one (which could look messier) or somehow redistribute the information to make the meaning behind the figure clearer.
- Figure 7c- Adding a legend on the figure would be useful and for consistency among other figure panels in this manuscript.
- Supplementary Figure 1b- It might also be important and useful to show the NMR spectra for the individual bases as controls (in absence of enzyme).
- Explanation of what a portal protein is should likely come early on in the manuscript rather than in the discussion. “Portal proteins such as YonE are required for pro-capsid assembly and correct DNA packaging and are thus essential for phage production (41)”.

Reviewer #4

(Remarks to the Author)

Version 1:

Reviewer comments:

Reviewer #1

(Remarks to the Author)

The authors have performed experiments to full address my concerns/comments. The manuscript can be accepted.

Reviewer #2

(Remarks to the Author)

I am satisfied with the authors response to my first comments. Thank you for the edits and thorough response, the updated cryo-EM figures look great.

Reviewer #3

(Remarks to the Author)

The authors have done more than expected during the interim and provide substantial new data supporting their claims and corroborating prior results. An already strong manuscript now feels even more robust. Might we suggest a few additional text edits stemming from the rebuttal discussion the authors have already expertly provided?

Likely the YonE:MBP-SbpK ratio that leads to maximal increase in NADase activity would be the one that matches the stoichiometry observed in SEC (10:1). The new titration data in Fig. R7 supports this idea and it could/should be stated more clearly in the text.

The authors should provide their rebuttal explanation for leaving out the no-protein controls in the NMR experiment in Supplementary Figure 1b in the main text, figure legend, or methods for completeness.

It would also round out the discussion about the stability of the ADPR molecules to include the authors rebuttal comments on this topic. Specifically regarding lyophilization.

The theory about metal ion impacts on observation of one or two ADPR species via LC-MS could be quickly tested for posterity. Add EDTA to see if it clears up the issue?

Reviewer #4

(Remarks to the Author)

Reviewer 1:

Comment: There are two major criticism that this reviewer had regarding the manuscript.

Response: We have addressed these criticisms in the comments below. We hope our responses help clarify these points and resolve the concerns raised.

Comment: The mechanistic details of the enzymatic reaction remains unclear despite the detailed cryoEM structure determination and the series of mutants used in this work. First, TIR domain is essential for enzymatic reaction of SPBK, which is clearly demonstrated in this work. However, the experimental design of the work is convoluted by polymerization of the TEV cleaved MBP-SPBK. While the MW of MBP-SPBK wild type and mutants D158R, N191R, E192A, W197A etc are similar (Figure 5), yet the enzymatic activity is present only for the TEV-cleaved WT. Does it imply that TEV-cleaved Mutants donot oligomerize ? Or does it oligomerize in a manner that is not conformationally competent for the reaction.

Response: In the MALS and mass photometry experiments, only the MBP-D158R mutant behave similar to wild-type MBP-SpbK. Both the wild-type protein and the D158R mutant exist as stable dimers. The N191R and W197A mutants exists as monomers in solution, while the G194R mutant forms both monomers and dimers. We did not make or test the E192A mutant. We have now included molecular weights in both the mass photometry and MALS panels in Figs. 3a-b, 5b and S4a-b to make it clearer that the molecular weights are not similar for wild type SpbK vs the N191R, P194R and W197A mutants. Our data demonstrate that the dimerisation defective N191R, P194R and W197A SpbK mutants cannot oligomerise into filaments. Hence there is strong correlation between NADase activity and ability to dimerise and oligomerise into filaments. D158R still retains the ability to oligomerise into filaments, and as the reviewer points out, the lack of NADase activity for this mutant is likely due to oligomerisation in a manner that is not conformationally competent for the NADase reaction (described on page 10 of the manuscript). In the revised manuscript, we have now more clearly highlighted that the D158R mutation likely promotes an alternate oligomerisation mode of SpbK:

“The D158R intrastrand BB-loop mutant formed stable dimers and assembled into filaments, although these were shorter than those of wild-type SpbK (Fig. 5a). This mutant also lacks NADase activity (Fig. 5c), indicating that the D158R substitution perturbs the intrastrand interface, resulting in an alternative mode of oligomerisation. This alteration likely compromises the structural integrity of the active site, which resides at the intrastrand BE interface, thereby abrogating its capacity to bind and hydrolyse NAD⁺.”

Comment: This reviewer strongly suggest (a) Structural studies of NAD⁺ (non hydrolyzable analogue) or (b) mutant protein that has intermediate activity or (c) constitutively active monomer/dimer. This is especially pertinent since the figure 3 demonstrates clearly that MBP-SPBK and TEV cleaved MBP_SPBK has drastically different kinetics.

Response: We do agree with the reviewer that structural studies with a non-hydrolysable NAD⁺ analog could be insightful, to understand in detail how the substrate engages with the active site. We did attempt to obtain a structure of SpbK with the non-cleavable NAD⁺ analog Carba-NAD⁺ which we synthesized in-house. Carba-NAD⁺ is identical to NAD⁺ except for the oxygen atom in the ribose ring, which is replaced by a carbon atom, forming a carbocyclic ring. We prepared Spbk filaments in the presence of 1 mM carbaNAD and collected ~5000 movies. After 2D classification, 8,791 filament segments were used for 3D reconstruction, yielding a 4.4 Å resolution reconstruction (Fig. R1 below) However, we did not observe electron density for Carba-NAD⁺ in the active site, suggesting that this NAD⁺ analog either binds only transiently or that the carbocyclic ring is not favourable for binding. As an alternative, we modelled the interaction between SpbK and NAD⁺ using AlphaFold 3 (Fig. 4g and S3e-g in manuscript). The results suggest that NAD⁺ engages the active site in a manner similar to the SARM1 and AbTIR 1AD/3AD complexes (Manik et al., 2022; Shi et al., 2022).

Unfortunately, we don't have a mutant with intermediate activity that can be used for structural studies. Considering that TIR NADase activity requires two TIR domains to come together to form an active site (Bhatt et al., 2024; Manik et al., 2022; Shi et al., 2022), we think it will be nearly impossible to design a constitutively active monomer, and because the active site is formed via an asymmetric head-to-tail interaction, designing a constitutively active dimer would also be extremely difficult and beyond the scope of this manuscript.

Comment: While oligomerization could be essential , Is it possible that the mode of polymerization influence the kinetics of the reaction?

Response: As discussed in detail above, our D158R mutagenesis data suggest that perturbation of the oligomerisation/polymerisation mode, specifically through the head-to-tail interface, modulates the kinetics of the reaction.

Fig. R1: CryoEM reconstruction of the SpbK filament in the presence of Carba-NAD⁺. (a) Flow-chart of the cryo-EM processing steps. (b) Gold-standard FSC curves of the final 3D reconstruction. (c) CryoEM map of active site region with fitted atomic model. The two TIR chains are highlighted in cyan and orange, respectively.

Comment: Second, the impact of YonE on the reaction mechanism of SPBK remains unclear to this reviewer. While the graph in The SEC profiles suggest that the two proteins interact, yet the relevance of this interaction in the NAD⁺ consumption is unclear to this reviewer. The graph is figure 7a and 3e suggests that kinetics of the control SPBK is different in two experiments at least in 16 hrs. Unless the mechanistic details of the reaction mechanism is clarified, the utility of Fig 8 remains inconsistent with the claims of this paper.

Response: We respectfully disagree with the reviewer's assessment that Fig. 7a does not demonstrate the relevance of the YonE:SpbK interaction in NAD⁺ consumption by SpbK. At the 16-hour time point, over 80% of NAD⁺ is hydrolysed when SpbK is incubated with YonE, compared to less than 5% in its absence. This difference provides strong evidence for a YonE-dependent enhancement of NAD⁺ consumption and is consistent with (Loyo et al., 2025), which demonstrated that NAD⁺ levels in *Bacillus subtilis* cells drop significantly only when SpbK and YonE are co-expressed. To further demonstrate the impact of YonE on the interaction, we have conducted additional kinetics experiments with different concentrations of YonE (0.2 μM, 1 μM and 2 μM) and updated Fig. 7a in the revised manuscript accordingly (Fig. R7 on page 11 below).

The experiments in Fig. 3e and Fig. 7a of the manuscript are not directly comparable, as we used different concentrations of MBP-SpbK and substrate. For the TEV cleavage experiments in Fig. 3e, we used 2 μM MBP-SpbK and 1 mM NAD⁺. For the YonE experiments in Fig. 7a we used 0.2 μM MBP-SpbK and 500 μM NAD⁺. While MBP fusion

markedly reduces SpbK oligomerization, at 2 μM , we suspect the concentration is sufficiently high to allow some degree of SpbK self-association, which may account for the low NADase activity. To make the results in Fig. 3e and Fig. 7a more directly comparable, we have repeated the SpbK +/-TEV NADase experiments in Fig. 3e using 500 μM NAD⁺ and two protein concentrations (0.2 and 2 μM) (Fig. R2 below).

Fig. R2: New Fig. 3e. Reaction progress curves of 0.2 μM MBP-SpbK +/-TEV protease + 500 μM NAD⁺ (left panel) and 2 μM MBP-SpbK +/-TEV protease + 500 μM NAD⁺ (right panel). MBP-SpbK was incubated with TEV protease for 1 h before addition of NAD⁺. The experiments were repeated two times with similar results.

Comment: While NMR has been used to monitor the enzymatic reaction of NAD consumption, an orthogonal method might be required, since there can be contribution from NH signals of the protein side chain.

Response: We understand reviewer #1 is concerned about the potential interference from protein NH signals towards our NMR-based enzymatic assays. We ensure that this should not be a concern as we have protein-only controls samples showing no such interference (see Fig. R3 below). The NMR assay is also well established, with several recent publications confirming its reliability (Bayless et al., 2025; Figley et al., 2021; Horsefield et al., 2019; Shi et al., 2022; Shi et al., 2024). Therefore, we believe that incorporating an additional NADase assay is unnecessary for this manuscript.

*: 1H NMR Peaks used for integration and quantification

Fig. R3: 1H NMR spectra of SpbK +/- NAD⁺.

Reviewer 2:

Comment: The work in this paper is significant because it illustrates yet another mechanism in the arsenal of bacterial defense systems. The authors successfully demonstrate that TIR-containing SpbK, when in the presence of the phage particle YonE, increases the NADase activity required to defend against pathenogenic activity.

Response: We thank the reviewer for the positive feedback highlighting the significance of our work.

Comment: The structural mechanism proposed could benefit from a few more experimental ideas: could cryoEM be performed on the stable dimer form of SpbK, and also, SpbK in complex with YonE? Both experiments could further illustrate the mechanistic story and may be simple to do since purification was already performed of each.

Response: The SpbK TIR domain dimer has a molecular mass of only 30 kDa; therefore we believe cryoEM would be extremely challenging if not impossible. The increased size of the fusion protein (monomer:~80 kDa; dimer ~160 kDa) is unlikely to help because MBP is connected to SpbK via a flexible linker and the linker between the N-terminal domain and the TIR domain is also predicted to be flexible (based on AlphaFold modelling and lack of density for MBP and the N-terminal domain in our filament reconstruction). Due to this flexibility, the MBP and N-terminal domains are likely to adopt many different orientations relative to each other and the TIR domain. This variability will prevent accurate alignment and averaging, which are essential for high-resolution maps.

We agree with the reviewer that a cryoEM structure of a YonE:SpbK complex could be informative. Unfortunately, it is not simple to do. We attempted to determine the cryoEM structure of YonE in complex with both SpbK^{Ntd} (Fig. R4, below) and the SpbK^{Ntd}-SARM1^{SAM} fusion protein (Fig. R5, below); however, we were unable to obtain high-quality 3D reconstructions.

To prepare the YonE:SpbK^{Ntd} complex for cryoEM, YonE (60 μM) was mixed with an excess of SpbK^{Ntd} (75 μM) and the mixture was subjected to analytical gel filtration, using a Superdex 200 column. Elution peak 1 contained both YonE and SpbK^{Ntd} and was used for cryoEM analysis. During cryoEM processing, we encountered similar challenges as observed with YonE alone, including preferred particle orientation and apparent flexibility in the stem and clip domains. Additionally, the gel filtration analysis revealed that only a small proportion of SpbK^{Ntd} forms a stable complex with YonE. The YonE:SpbK^{Ntd} peak likely comprises a mixture of free YonE and YonE:SpbK^{Ntd} complexes, further complicating cryoEM data processing, particularly due to the limited range of particle views. As with YonE alone, the reconstruction yielded interpretable density only for the wing and crown domains of YonE, providing no structural insight into the interaction between SpbK^{Ntd} and YonE.

Fig. R4: CryoEM reconstruction of the YonE:SpbK^{Ntd} complex. (a) Gel filtration analysis (b) SDS-PAGE analysis of gel filtration fractions containing the YonE:SpbK^{Ntd} complex (c) Negative-stain electron micrograph. (d) Flow-chart of the cryo-EM processing steps. Additional 3D classification (not shown in this figure), did not identify any classes displaying density for the YonE:SpbK^{Ntd} complex.

To prepare the YonE:S SpbK^{Ntd}-SARM1^{SAM} complex for cryoEM, YonE (25 μM) was mixed with SpbK^{Ntd}-SARM1^{SAM} (25 μM) and the mixture was subjected to analytical gel filtration using a Superdex 200 column. Elution peak 1 contained both YonE and YonE: SpbK^{Ntd}-SARM1^{SAM} and was used for cryoEM analysis. To characterise the complex formed by YonE and the SpbK^{Ntd}-SARM1^{SAM} fusion protein, we first collected negative stain EM images from gel filtration peak fractions. The initial portion of the peak appeared heterogeneous, with numerous large aggregates. We suspect that fusion of SpbK^{Ntd} to the octameric SARM1^{SAM} domain ring enables multivalent interactions with YonE dodecamers, promoting aggregation. The later portion of the gel filtration peak was more homogeneous, predominantly containing ring-shaped particles consistent in size with either YonE dodecamers or SARM1^{SAM} octamers. We focused on this fraction for cryoEM analysis. Unfortunately, from a cryoEM dataset comprising ~4,400 movies, we were only able to obtain two single well-defined 'side-view' 2D classes containing ~14,000 particles (Fig. R5), which was insufficient for 3D reconstruction. We suspect that multivalent interactions persist in the sample, complicating single-particle analysis. The side-view

classes revealed additional density beneath the stem/clip region of YonE, which we believe corresponds to the octameric SARM1^{SAM} ring.

Fig. R5: (a) Gel filtration analysis. (b) SDS-PAGE analysis of gel-filtration fractions. The chromatogram and gel in (a-b) are from Fig. 7a in manuscript. (c) Negative-stain electron micrographs of the YonE:SpbK^{Ntd}-SARM1^{SAM} gel filtration peak 1. Left and right panels correspond to the first and second halves of the gel filtration peak 1, respectively. (b) Representative 2D class averages from cryoEM processing of YonE:SpbK^{Ntd}-SARM1^{SAM} in CryoSPARC.

Comment: Also, for the cryoEM reconstruction of YonE, I am curious to see the full computational processing workflow. Was any symmetry or 3D classification performed prior to the final reconstruction of this 12-mer density of YonE? If so, it would be ideal to inspect the C1 solution prior to any imposed symmetry, as well as any other resulting 3D classes, to see if the missing base units can be found. This could be included in the supplemental processing Fig., but it would lend to a better understanding or interpretation of the missing density, which can potentially get averaged out if it does not follow an imposed symmetry rule.

Response: We agree with the reviewer that the density corresponding to the stalk and clip domain of YonE may be averaged out if it does not conform to the imposed C12 symmetry. In addition to the workflow shown in Fig. S5, we also performed non-uniform refinement using C1 symmetry (~4.6 Å resolution), as well as 3D classification. However, inspection of the resulting maps did not reveal any density for the missing regions (see Fig. R6 below), indicating that the absence of density is unlikely to be a symmetry-related issue.

We have updated Supplementary Fig. S5 to include the C1 symmetry processing step. The 3D classification results were not included, as none of the resulting classes were used for further analysis.

Fig. R6: Additional cryoEM processing steps for YonE. a) Non-uniform refinement, C1 symmetry. b) 3D classification, C1 symmetry, 10 classes.

Comment: Overall the findings do support and elucidate SpB mechanism in the fight against SpB phage infection, and it adds to the expansive set of tools available to bacterial immune systems.

Response: We thank the reviewer for this positive assessment.

Reviewer #3:

Comment: SpbK is a TIR domain containing protein from Bacillus, shown previously to be an enzyme with NAD⁺ depletion activity which presumably leads to cellular growth arrest in response to phage infection. The authors have elucidated the structure of higher order filaments using cryo-EM and established structural similarity of these filaments to other known TIR-domain containing proteins. In addition, the authors describe interactions between YonE (a portal protein from phage SPbeta) and SpbK. Based on separate structural determinations of SpbK and YonE complemented with Alphafold predictions and mutagenesis experiments, the authors have determined residues/interfaces (namely, SpbK's N terminal domain to YonE's clip domain) which are key to interactions between SpbK and YonE.

The authors have presented us with a very thorough study that has layers of details which will of interest to biochemists, structural biologists, phage biologists, and those studying immune signaling. This represents one of the few research articles in recent memory that are explicit about the pitfalls of studying cyclic ADPR analogs and that provide clear and in-depth analysis of these molecules. The story is clear and the data and analysis are original and exciting. Minor comments mainly on the formatting of figures and suggestions for one or two additional experiments to solidify alphafold predictions can be found below.

Response: We thank the reviewer for the positive feedback and constructive suggestions.

Comment: First results subheading “an” NADase not “a”

Response: We thank the reviewer for the suggestion. The subheading has been updated in the revised manuscript.

Comment: For the double-peaks observed for N1-cADPR: Can the authors speculate more on what component in lysate could be causing this phenomenon? Metals perhaps? Could the second peak which disappears following pre-purification be a less-stable isomer which simply does not survive the lyophilization procedure?

Response: Indeed, a metal cation is our leading hypothesis, because the phosphates may bind it tightly, which could stabilise cADPR into different conformational isomers. The latter hypothesis is unlikely because we have lyophilized the sample and redissolved it (without a prior separation), and we still see both peaks; in other words, lyophilisation alone does not seem to make one of the peaks disappear.

We added a parenthetical statement about divalent metal cations to the revised manuscript (page 6):

“We hypothesised that either two cADPR variants were produced by the cell, or components in the cell lysate (e.g., divalent metal cations) caused the N1-cADPR to adopt different configurations with different retention times.”

Comment: When discussing alphafold prediction of the hexamer bound to NAD⁺, the authors should consider adding mutagenesis data to supplement the supposition. Mutants of residues suspected to form the composite active site are especially interesting (perhaps K224 stands out as the authors mention this residue specifically).

The composite active site has been observed across multiple bacterial, plant and animal/human TIR NADase structures (Bhatt et al., 2024) and has already been extensively validated through site-directed mutagenesis in both bacterial (AbTir) and human (SARM1) TIR domains (Manik et al., 2022; Shi et al., 2022). Loyo and Grossman demonstrated that E192, which interacts with the nicotinamide ribose, is essential for SpbK NADase activity (Loyo and Grossman, 2025). Importantly, our structural model predicts that D158 plays a key role in forming the composite active site, and we have shown that its mutation completely abolishes NADase activity. As such, we consider additional mutagenesis data for the predicted SpbK:NAD⁺ interaction unlikely to yield substantial new insights.

Furthermore, structural studies of both AbTIR and SARM1 bound to NAD mimetics have shown that the adenine ring forms key interactions with the protein backbone. Thus, mutations of side chains such as K224 may have limited impact on SpbK NADase activity. This is exemplified by SARM1, where mutation of W662 (W662A), whose side chain stacks against the adenine ring, results in only a modest reduction in NADase activity (Shi et al., 2022).

Comment: Can the authors please comment here and in the main text why they chose to remove the first 45 amino acids of the YonE construct used for cryo-EM analysis? The statement that the residues were truncated is made but no justification is provided.

Response: We removed the first 45 first amino acids of YonE in the expression construct as they are predicted to be disordered. We have included the following sentence in the main text of revised manuscript:

We expressed and purified a truncated YonE construct lacking the first 45 residues predicted to be disordered (Jumper et al., 2021).

Comment: In experiments mixing YonE and SpbK WT/mutants, why were those concentrations tested? (2uM and 0.2 uM, respectively)? If this is simply due to the expected stoichiometry of the final complex, then the authors should amend the manuscript with this detail for completeness and/or conduct additional experiment - titration of [YonE] into SpbK.

Response: The ratio of 2 μ M YonE to 0.2 μ M MBP-SpbK was not selected based on stoichiometry, but rather on the significant enhancement in NADase activity observed relative to MBP-SpbK alone. As suggested by the reviewer, we have conducted additional experiments with different concentrations of YonE (0.2 μ M, 1 μ M and 2 μ M) and updated Fig. 7a in revised manuscript accordingly (Fig. R7 below).

Fig. R7: New Fig. 7a. Reaction progression curves for 0.2 μM MBP-SpbK + 0.2-2 μM YonE. The initial NAD⁺ concentration was 500 μM. The experiments were performed two times with similar results.

Comment: Can the authors share the SpbK NTD to YonE monomer to monomer alphafold prediction (possibly as a supplementary figure)? The statistics might look better and still provide useful information regarding the binding interactions.

Response: We have included a figure below, showing the AlphaFold 3 models of the monomeric SpbK^{NTd}:YonE^{Clip} complex. The predicted ipTM scores are substantially lower than those observed for the dodecameric complex (ipTM = 0.73), and there is no consistency among the five predicted models (Fig. R8a). Importantly, none of these models recapitulate the interactions seen in the dodecameric complex (Fig. R8b). One possible explanation is that multiple interaction interfaces are required for stable complex formation. Specifically, in the model of the dodecameric complex, the α_3 - α_4 loop of SpbK^{NTd} docks onto the YonE^{Clip}:YonE^{Clip} interface, consistent with our mutagenesis data. The dodecameric complex model also suggests that SpbK^{NTd}:SpbK^{NTd} interactions are necessary for assembly, which is also supported by our mutagenesis data.

Fig. R8: AlphaFold 3 models (M1-M5) of the monomeric YonE-SpbK^{Ntd} complex. Only the clip domain region of YonE (residues 270-225) was used for modelling. (a) Monomeric YonE-SpbK^{Ntd} complex models, colored by the confidence metric pLDDT (predicted local distance difference test). (b) Monomeric YonE-SpbK^{Ntd} complex models superimposed onto the dodecameric YonE-SpbK^{Ntd} complex model. The AlphaFold3 models were superimposed using the YonE clip domain.

Comment: Figures with SEC-MALS should clearly describe in the figure legends what is indicated by the orange trace observed in the graphs.

Response: We have added the following sentence to the legend of Fig. 3b, 5b and S4b “The blue line represents the refractive index trace, while the orange line represents the average molecular mass distribution across the peak.”

Comment: Figure 5f and 5g- The wildtype *spbK* + *yonE* trace needs to be presented as well for appropriate comparison and indeed it might be beneficial to combine both panels into one (which could look messier) or somehow redistribute the information to make the meaning behind the figure clearer.

Response: In response to Reviewer #2's suggestion, we have added the wild-type *spbK* traces and wild-type *spbK* + *yonE* from (Loyo and Grossman, 2025) to enable appropriate comparison (see Fig. R9 below). We believe that the inclusion of these controls improves the clarity of the data presented. However, we feel that combining the figures would make them more cluttered and harder to interpret.

To improve clarity further, we expanded the description of these figures in the revised manuscript:

“All of the mutants exhibited growth comparable to wild-type *spbK* (Fig. 5f), and consistent with their loss of NADase activity, they did not induce growth arrest in the presence of YonE (Fig. 5g).”

Fig. R9: New Fig. 5f-g. (f) Growth curves of strains expressing various alleles of *spbK*. (g) Growth curves of strains expressing both *yonE* and various alleles of *spbK*. 1 mM IPTG was used to induce expression of *yonE*. IPTG was added directly after measuring OD600 at time 0. Cells expressing *yonE* and *spbK* mutants do not exhibit the growth arrest phenotype observed when *yonE* is expressed with wild-type *spbK*. *spbK* alleles are expressed under the native promoter of *spbK*. Data shown in f and g are from three biological replicates. Error bars represent standard deviations. The wild-type *spbK* + *yonE* and wild-type *spbK* traces in (f-g) are from Loyo and Grossman, 2025 (Loyo and Grossman, 2025).

Comment: Figure 7c- Adding a legend on the figure would be useful and for consistency among other figure panels in this manuscript.

Response: We have added a legend to this figure in the revised manuscript.

Comment: Supplementary figure 1b- It might also be important and useful to show the NMR spectra for the individual bases as controls (in absence of enzyme).

Response: We understand reviewer #3’s concern about the absence of no-protein control spectra. However, these bases do not show NMR signals in this anomeric region (5.4 to 6.4 ppm) and thus we believe it would be easier for readers to understand without no-protein controls. We have recently published a similar figure (Shi et al., 2024).

Comment: Explanation of what a portal protein is should likely come early on in the manuscript rather than in the discussion. “Portal proteins such as YonE are required for pro-capsid assembly and correct DNA packaging and are thus essential for phage production (41)”.

Response: We thank the reviewer for this suggestion. We have moved this sentence to the introduction (page 4, paragraph 2) in the revised manuscript.

References

- Bayless, A.M., *et al.* (2025). The Arabidopsis TIRome informs the design of artificial TIR (Toll/interleukin-1 receptor) domain proteins. *Proc Natl Acad Sci U S A* 122, e2505893122.
- Bhatt, A., Mishra, B.P., Gu, W., Sorbello, M., Xu, H., Ve, T., and Kobe, B. (2024). Structural characterization of TIR-domain signalosomes through a combination of structural biology approaches. *IUCrJ* 11, 695-707.
- Figley, M.D., *et al.* (2021). SARM1 is a metabolic sensor activated by an increased NMN/NAD⁺ ratio to trigger axon degeneration. *Neuron*.
- Horsefield, S., *et al.* (2019). NAD(+) cleavage activity by animal and plant TIR domains in cell death pathways. *Science* 365, 793-799.
- Jumper, J., *et al.* (2021). Highly accurate protein structure prediction with AlphaFold. *Nature* 596, 583-589.
- Loyo, C.L., and Grossman, A.D. (2025). A phage-encoded counter-defense inhibits an NAD-degrading anti-phage defense system. *PLoS Genet* 21, e1011551.
- Manik, M.K., *et al.* (2022). Cyclic ADP ribose isomers: Production, chemical structures, and immune signaling. *Science* 377, eadc8969.
- Shi, Y., *et al.* (2022). Structural basis of SARM1 activation, substrate recognition, and inhibition by small molecules. *Mol Cell* 82, 1643-1659 e1610.
- Shi, Y., *et al.* (2024). Structural characterization of macro domain-containing Thoeris antiphage defense systems. *Sci Adv* 10, eadn3310.

Response to reviewer comments:

Reviewer #1

The authors have performed experiments to full address my concerns/comments. The manuscript can be accepted.

Response: We appreciate the reviewer's recommendation for acceptance and are glad the revisions addressed their concerns

Reviewer #2

I am satisfied with the authors response to my first comments. Thank you for the edits and thorough response, the updated cryo-EM figures look great.

Response: We appreciate the reviewer's positive feedback and are pleased that the revisions and updated cryo-EM figures meet their expectations.

Reviewer #3

The authors have done more than expected during the interim and provide substantial new data supporting their claims and corroborating prior results. An already strong manuscript now feels even more robust. Might we suggest a few additional text edits stemming from the rebuttal discussion the authors have already expertly provided?

Response: We appreciate the reviewer's positive assessment and provide our responses to the suggested minor edits below.

Likely the YonE:MBP-SbpK ratio that leads to maximal increase in NADase activity would be the one that matches the stoichiometry observed in SEC (10:1). The new titration data in Fig. R7 supports this idea and it could/should be stated more clearly in the text.

Response: We used different SpbK variants in our experiments: the enzyme assays employed the full-length protein fused to MBP, whereas size-exclusion chromatography utilised constructs lacking the TIR domain (either the N-terminal domain alone or fused to GST or SARM1^{SAM}). MBP likely influences SpbK's ability to self-associate (see Fig. 2 in the manuscript). Additionally, initiation of TIR domain oligomerisation by YonE may stabilise the SpbK:YonE interaction and affect stoichiometry. Given these complexities, we prefer not to draw direct correlations between these experiments in the main text.

The authors should provide their rebuttal explanation for leaving out the no-protein controls in the NMR experiment in Supplementary Figure 1b in the main text, figure legend, or methods for completeness.

Response: We have now included the reason 'for leaving out the no-protein controls in the NMR experiment in the Supplementary Figure 1b legend: Spectra corresponding to no-protein samples are not shown as anomeric peaks for base-exchange products are evident by comparing with the control (bottom).

It would also round out the discussion about the stability of the ADPR molecules to include the authors rebuttal comments on this topic. Specifically regarding lyophilization.

Response: Addressing the impact of lyophilisation on N1-cADPR stability with sufficient rigor would require additional replicates and experimental work. We believe such effort would not add to the conclusions of this paper, and since the rebuttal will be published alongside the manuscript, we prefer to retain the lyophilisation discussion in the rebuttal rather than incorporating it into the main text.

The theory about metal ion impacts on observation of one or two ADPR species via LC-MS could be quickly tested for posterity. Add EDTA to see if it clears up the issue?

Response: We appreciate this suggestion and agree that metal ion interactions could be an interesting aspect to explore. However, implementing this test would require generating new lysates and additional LC-MS runs, which represents a considerable amount of work. Since this experiment is not essential for the conclusions of this study, we prefer to leave it for future investigations.